# Causal Graphical Models for Vision-Language Compositional Understanding

**Fiorenzo Parascandolo, Nicholas Moratelli, Enver Sangineto, Lorenzo Baraldi & Rita Cucchiara**
AImageLab, Dipartimento di Ingegneria "Enzo Ferrari"
University of Modena and Reggio Emilia
`name.surname@unimore.it`

## Abstract

Recent work has empirically shown that Vision-Language Models (VLMs) struggle to fully understand the compositional properties of the human language, usually modeling an image caption as a "bag of words". As a result, they perform poorly on compositional tasks, which require a deeper understanding of the different entities of a sentence (subject, verb, etc.) jointly with their mutual relationships in order to be solved. In this paper, we model the dependency relations among textual and visual tokens using a *Causal Graphical Model* (CGM), built using a *dependency parser*, and we train a decoder conditioned by the VLM visual encoder. Differently from standard autoregressive or parallel predictions, our decoder's generative process is partially-ordered following the CGM structure. This structure encourages the decoder to learn only the main causal dependencies in a sentence discarding spurious correlations. Using extensive experiments on five compositional benchmarks, we show that our method significantly outperforms all the state-of-the-art compositional approaches by a large margin, and it also improves over methods trained using much larger datasets. Our model weights and code are publicly available.[1]

## 1 Introduction

Vision-Language Models (VLMs) have shown impressive results in different tasks such as, for instance, zero-shot classification, image-text retrieval, vision-question answering, image-captioning, and many others (Radford et al., 2021; Li et al., 2023b; Singh et al., 2021; Liu et al., 2023). However, despite this success, most VLMs still struggle in understanding the compositional nature of the human language. For instance, Yuksekgonul et al. (2023) empirically showed that common VLMs usually do not consider the order and the syntactic/semantic relations of words in a sentence, which is treated as a *bag of words*, where "the horse is eating the grass" and "the grass is eating the horse" can easily be confused. Jointly with Yuksekgonul et al. (2023), many other authors have recently proposed different *compositional benchmarks* which confirm the poor performance of common VLMs when tested against compositional tasks (Hsieh et al., 2024; Zhao et al., 2022; Burapacheep et al., 2024; Peng et al., 2024). One of the probable reasons of this bag-of-words behavior is the contrastive loss used in CLIP (Radford et al., 2021) (and in other VLMs), which compares a single vector representing the textual encoder's output with a single vector representing the visual encoder's output, sacrificing textual and visual details (Yuksekgonul et al., 2023; Kamath et al., 2023; Basu et al., 2024). Another reason is the low quality of the captions used for VLM pre-training, which are usually noisy or do not describe the details of the image and the interactions among its objects (Doveh et al., 2023a).

Most of the compositional methods that have been recently proposed to alleviate this problem focus on creating annotations with a richer compositional structure, used to fine-tune a VLM (Yuksekgonul et al., 2023; Doveh et al., 2023a; Cascante-Bonilla et al., 2023). For instance, NegCLIP (Yuksekgonul et al., 2023) creates *hard negatives*, in which the original caption is modified swapping the positions of some words, and these hard negatives are used jointly with common negatives to fine-tune CLIP using the standard contrastive loss. However, the automatic creation of hard negatives is

---

itself noisy, leading to captions which often do not have a correct syntactic/semantic meaning (this problem is inherited by some compositional benchmarks, see Sec. 4). In (Tschannen et al., 2023), a VLM is pre-trained from scratch using a captioning strategy and a huge private dataset. Specifically, the authors propose both Cap, where the pre-training strategy is a standard *AutoRegressive* (AR) next-token prediction, and CapPa, where the AR training is mixed with a *parallel* training (Bolelli et al., 2018), in which all the textual tokens are simultaneously predicted. Tschannen et al. (2023) show that both Cap and CapPa achieve excellent results on compositional tasks, and argue that a *generative training* encourages the VLM to focus on fine-grained descriptions of the visual content.

In this paper, inspired by Cap and CapPa, we propose a VLM adaptation approach for compositional reasoning which is based on a decoder trained with a captioning strategy. However, differently from the standard fully-sequential AR and the parallel predictions used in (Tschannen et al., 2023), we propose a partially ordered, semi-parallel AR prediction strategy which is guided by the dependency relations of a *Causal Graphical Model* (CGM) (Schölkopf et al., 2021). In more detail, we use an off-the-shelf *dependency parser* (Dozat & Manning, 2016), which creates a syntactic tree from a given textual sentence. Specifically, given a caption, a dependency parser automatically builds a *Dependency Tree* (DT), in which each node is associated with a caption word and each edge represents a syntactic dependency relation between two words (Fig. 1). The DT, jointly with the visual features extracted from the image using a frozen visual encoder, are used to build a CGM, which describes the dependency relations among image patches and textual tokens. Our token prediction strategy is based on the dependency relations contained in this CGM. The rationale behind this approach is illustrated in Fig. 1 using the caption "A brown bird has a small yellow head". For instance, in the resulting DT, the adjective "brown" depends on the noun "bird". However, using a standard AR approach, where the token prediction order follows the English grammar, the captioning model should predict "brown" before knowing that this adjective refers to "bird", which is a quite ambiguous task, since many objects may be brown in the image. Conversely, when our model predicts the adjective ("brown"), it knows the noun ("bird") it refers to, thus the word generation can be specific to the entities, the attributes and the relations contained in the input image. Generally speaking, we factorize the joint distribution of all the caption words following the *disentangled factorization* of a CGM (Schölkopf et al., 2021), and our semi-parallel AR model predicts a token conditioned only on the tokens on which it depends. For instance, in the example of Fig. 1, "small" and "yellow" are predicted in parallel and they are conditionally independent given "head", thus no statistical dependence is learned between these two words. The advantage of this strategy is that the decoder can focus on learning only the main causal dependency relations, ignoring possible spurious associations (Pearl & Verma, 1995) induced by the sequential order of the words in a natural language sentence. Moreover, we use the same prediction strategy also at inference time, when we compute the likelihood of a candidate caption. In this case too, the use of the CGM makes the likelihood estimation independent of spurious associations due to the sequential order of the words.

We validate our method using different VLMs (CLIP, XVLM (Zeng et al., 2022) and InstructBLIP (Dai et al., 2023)). Using extensive experiments with five compositional datasets, we show that our approach largely outperforms all previous works, setting a new *state of the art* in all the evaluated benchmarks, and that it also improves on Cap and CapPa, despite being trained on much less data.

## 2 RELATED WORK

**Compositional Methods.** Most of the compositional methods are based on creating *annotated training samples* which force the VLM to acquire compositional knowledge. For instance, (Yuksekgonul et al., 2023; Zhang et al., 2024; Huang et al., 2024; Buettner & Kovashka, 2024; Momeni et al., 2023; Doveh et al., 2023b; Singh et al., 2023; Oh et al., 2024; Yellinek et al., 2023; Herzig et al., 2023) use either a rule-based method or a Large Language Model (LLM) to create *hard negatives* (Sec. 1), which typically consist in replacing or swapping the position of some words in the ground-truth caption associated with a training image. In (Cascante-Bonilla et al., 2023), dense captions are constructed using synthetic videos created with a 3D physics-based simulator, while (Singh et al., 2024) use real videos. In DAC (Doveh et al., 2023a), dense captions are created by combining the results of either an LLM (GPT-NEO-2.7B) or a segmentation network (SAM (Kirillov et al., 2023)) with a captioner (BLIP-2 (Li et al., 2023b)). SAM is also used in (Sahin et al., 2024) jointly with Stable Diffusion (Rombach et al., 2022) to generate hard negative *images*. Moreover, Stable Diffusion is used in (Li et al., 2023a; Clark & Jaini, 2023; Krojer et al., 2023) as an alternative VLM. The

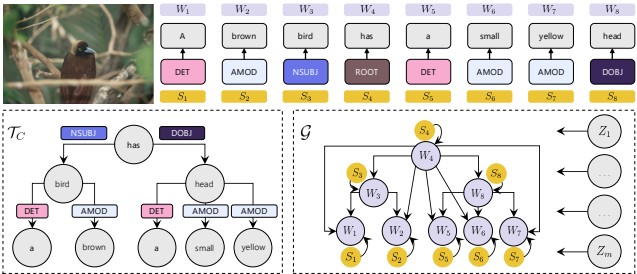

Figure 1: Dependency relations between words in a sentence. On the left, the DT ($\mathcal{T}$) extracted from the caption shown above using a dependency parser (Dozat & Manning, 2016). On the right, the corresponding CGM ($\mathcal{G}$). To improve readability, in $\mathcal{G}$ we use different colors for different variable types and we omit the causal dependencies between visual ($Z$) and textual ($W$) variables.

main idea is that the noise prediction error of the Diffusion Model (DM) (Ho et al., 2020), obtained by feeding Stable Diffusion with a corrupted version of the test image and a given caption, can be used as an estimate of the image-caption similarity. Finally, Stable Diffusion is used in (Basu et al., 2024) as an additional regularization loss to fine-tune CLIP.

Apart from DM-based methods, most of the compositional approaches are based on fine-tuning or adapting CLIP. For instance, Zhang et al. (2024) use an hinge loss with a curriculum-learning based adaptive margin, while Doveh et al. (2023a) use a Multiple Instance Learning loss. Curriculum learning is used also in (Singh et al., 2023), while Zheng et al. (2024) iteratively retrain CLIP and represent an image using a sparse combination of codebook codes. Oh et al. (2024) propose a local hard negative loss to fine-tune CLIP which is based on a dense alignment betweeen patch embeddings and textual token embeddings. Wazni et al. (2024) use a dependency parser (see below) to extract triplets (subject, verb, object) from a caption. Subjects and objects are represented as embedding vectors using the CLIP textual encoder, while verbs are represented by matrices that are multiplied with either the subject or the verb to change their meaning. In (Li et al., 2024a), CLIP is embedded in a larger VLM, which includes a detection network and an LLM. The LLM comunicates with the detection network using special tokens. A few methods use *generative pre-trained* VLMs, and their results usually show a large improvement with respect to encoder-based VLMs when applied to compositional tasks, most likely because the next-token prediction pre-training encourages the VLM to learn the natural language compositional characteristics. For instance, Herzig et al. (2023) use "Adaptive Scene Graph Tokens" to adapt both CLIP and BLIP-2 (Li et al., 2023b) to predict scene graph information, and they show that the BLIP-2 based results are much higher than those based on CLIP. Wan et al. (2024) use LLaVA (Liu et al., 2023) and a classifier-free guidance strategy, in which they compare the VLM prediction on two images: the original test image and a modified version where the main objects are masked-out. BLIP is used also by (Lin et al., 2024), who focus on mitigating the linguistic bias on the VLM pre-training dataset. Finally, Tschannen et al. (2023) propose two VLMs, called Cap and CapPa (see Sec. 1), both trained generatively. Cap is a standard AR captioner, while CapPa is trained using a combination of 25% AR next-token prediction and 75% fully-parallel token prediction. Inspired by the success of generative pre-training, in this paper we propose a decoder trained using a semi-parallel prediction strategy, where the order in which future tokens are predicted depends on a CGM, and we show that it can be applied to both encoder-only and generative pre-trained VLMs, significatively boosting the results of both.

**Causal Graphical Models.** CGMs are used in causal learning to represent the causal relations among a set of variables (Schölkopf et al., 2021; Perry et al., 2022). These relations are supposed to be known and are represented by the edges connecting each variable (node) in the graph with the variables on which it depends ("parents"). The joint distribution of all the variables of the CGM is computed using the *disentangled factorization* (Schölkopf et al., 2021; Perry et al., 2022), given by the product of all the conditional distributions of each variable with respect to its parents (App. A). The main advantage of this factorization is that, since it is assumed to be *causally sufficient*, the model does not need to learn other inter-variable conditional distributions, in this way reducing the number of training samples necessary to learn the joint distribution (Schölkopf et al., 2021). As far as we know, the only work using CGMs for vision-language compositional tasks is (Jiang et al., 2024), where *Independent Causal Mechanisms (ICMs)* (Parascandolo et al., 2018; Goyal et al., 2021;

Schölkopf et al., 2021) describe the relations between the subject, the object and the action of an image. However, the method proposed in (Jiang et al., 2024) is radically different from our proposal, being each ICM simply computed as the CLIP similarity between a word and a sub-image.

**Syntactic Trees.** In *Dependency Grammars*, dependency relations are syntactic and semantic connections between words in a sentence, where one word (called "head") governs or determines the grammatical behavior of the "dependent" word (Nivre, 2005). Given a sentence, these dependencies are organized in a Dependency Tree (DT), which can be automatically extracted using a parser (Honnibal et al., 2020; Zhang et al., 2020; Dozat & Manning, 2016). In (Yang & Wan, 2022) a Language Model (LM) is trained to predict whether the future tokens in the sequence are head or dependent of a previously observed token, and this prediction replaces the standard Maximum Likelihood Estimation objective. Similarly, in (Deguchi et al., 2019) the dependency relations extracted by an external parser are learned by the LM and used to modulate the Transformer (Vaswani et al., 2017) attention maps. A DT can also be used to compute a *syntactic distance* between words, which in turn can be used, e.g., as an additional loss (Du et al., 2020) or to modulate the attention maps (Hou et al., 2022). In the vision-language domain, DTs are used in (Song et al., 2022) to convert a textual question into a template for CLIP fine-tuning, and in (Li et al., 2024b) to replace and swap words in a sentence and fine-tune a VLM using the correction of the modified sequence as a pretext task. Finally, *constituency parsers* group words that belong to a specific grammatical category in a sub-phrase. Constituency trees are used, e.g., in (Yellinek et al., 2023) to generate sub-phrase specific hard negative captions or in (Zhang et al., 2022) to extended the contrastive loss by maximizing the similarity of words in the same sub-phrase. Differently from previous work, we use a DT to extract syntactic and semantic dependencies between the words of a sentence, and we interpret these dependencies as causal relations that guide the construction of our CGM.

## 3 METHOD

Given an image-caption pair $(X, C)$, our goal is to define a set of conditional distributions over the random variables associated with the image features and the caption words. For this purpose, as anticipated in Sec. 1 and 2, we use an off-the-shelf dependency parser (Dozat & Manning, 2016) which, for a specific $C = [w_1, ..., w_n]$, returns a DT $\mathcal{T}^2$ (Fig. 1), where each node corresponds to a word and each edge $(i, j)$ connects the "dependent" word $w_j$ with its "head" $w_i$ (Sec. 2). $\mathcal{T}$ contains the syntactic and semantic dependencies between the words in $C$ (Nivre, 2005), and we make this dependency explicit by connecting each word to all the words it transitively depends on in the tree. Specifically, we define a CGM $\mathcal{G}$ by associating each word $w_j$ with a random variable $W_j$, corresponding to a node of $\mathcal{G}$. Moreover, we connect the node corresponding to $W_j$ with all the variables corresponding to the ancestors of $w_j$ in $\mathcal{T}$ (Fig. 1). Formally, if $w_{i_1}, ..., w_{i_k}$ are the ancestors of $w_j$ in $\mathcal{T}$, then we assume a causal dependence between the corresponding variables: $W_{i_1} \to W_j, ..., W_{i_k} \to W_j$. Furthermore, the parser labels each word in $\mathcal{T}$ with a syntactic type using a prefixed vocabulary $V$ (Silveira et al., 2014; Zhang et al., 2020). For instance, if $type(w_j) = $ nsubj $\in V$, it means that $w_j$ is a noun and it plays the role of the subject in the sentence. Intuitively, we can think of these syntactic types as categorical syntactic features extracted from $C$, which we formally describe using $n$ random variables $S_1, ..., S_n$, where each $S_j$ ranges over $V$. In $\mathcal{G}$, we assume that each $W_j$ depends on its corresponding syntactic variable $S_j$: $S_j \to W_j$.

Finally, we extract a set of features from $X$ using the VLM visual encoder $\mathcal{E}$: $\mathcal{Z} = \mathcal{E}(X) = \{z_1, ..., z_m\}$ (details in Sec. 3.1), and, similarly to the textual case, we associate a random variable $Z_k$ to each feature $z_k \in \mathcal{Z}$. In $\mathcal{G}$, we assume that $W_j$ depends on all the visual variables: $Z_1 \to W_j, ..., Z_m \to W_j$. Using the above assumptions, we define the *parents* (Schölkopf et al., 2021) of $W_j$ as: $\mathbf{PA}(W_j) = \{W_{i_1}, ..., W_{i_k}, S_j, Z_1, ..., Z_m\}$, and we model the conditional joint distribution of the textual variables given the visual and the syntactic variables as:

$$P(W_1, ..., W_n | S_1, ..., S_n, Z_1, ..., Z_m) = \prod_{j=1}^{n} P(W_j | \mathbf{PA}(W_j)), \quad (1)$$

where the right side of Eq. (1) is obtained using the *disentangled factorization* of CGMs (Schölkopf et al., 2021; Perry et al., 2022) and assuming that $S_1, ..., S_n$ and $Z_1, ..., Z_m$ are independent of each

---

[2]Note that, for each $C$ in the training/testing set, $\mathcal{T}$ needs to be extracted only once and can be done offline.

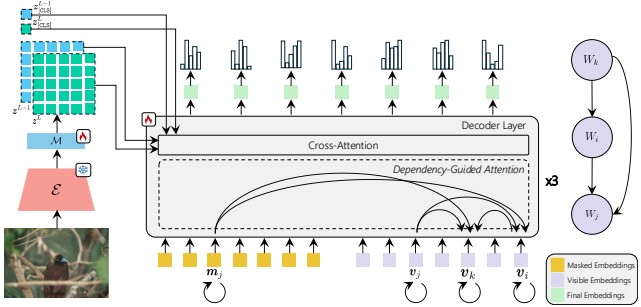

Figure 2: A schematic illustration of our decoder.

other (see App. A for more details). In Sec. 3.1 we show how a VLM can be adapted to predict this disentangled factorization both at training and at inference time.

**Discussion.** Tschannen et al. (2023) formulate the joint distribution of the words in $C$ using the standard AR prediction strategy commonly adopted by image captioning methods (Sec. 1):

$$P(W_1, ..., W_n | Z_1, ..., Z_m) = \prod_{j=1}^{n} P(W_j | W_1, ..., W_{j-1}, Z_1, ..., Z_m). \tag{2}$$

The advantage of our formulation (Eq. (1)) over Eq. (2) is that, in our case, the model needs to learn only the inter-variable conditional distributions indicated by the dependency parser, reducing the risk of overfitting on the training data (Schölkopf et al., 2021). Specifically, the dependency parser helps in discarding those spurious *associations* (Pearl, 2009; Pearl & Verma, 1995) contained in Eq. (2) which depend on the sequence of words in $C$ but do not correspond to a strict semantic/syntactic relation (e.g., "small" and "yellow" in the example of Sec. 1). In contrast, we interpret the dependency relations extracted by a dependency parser as *causal relations* because they directly model the (linguistic) influence of the "head" variable with respect to the generation of the "dependent" variable. For instance, the probability values of an adjective are directly influenced by the noun it refers to, because the adjective describes an attribute *of that noun*, thus the corresponding conditional probability is not a spurious association (more details in App. A).

While the causal dependency relations in $C$ may not be exhaustively described by $\mathcal{G}$ and there may be other relations between words in $C$, we follow (Goyal & Bengio, 2020), and we assume that, in a symbolic domain like the natural language, the joint distribution over the words of a sentence should be *sparse*. This is also in line with very recent work which shows that sparse attention in Transformers helps the network focus on the most relevant context and improves its performance removing noise (Leviathan et al., 2024; Ye et al., 2024). Thus, we prefer sparseness to completeness and we assume that the word dependencies extracted by a dependency parser are causally sufficient (Sec. 2, with more details in App. A). Finally, Tschannen et al. (2023) propose also a parallel prediction strategy, which corresponds to:

$$P(W_1, ..., W_n | Z_1, ..., Z_m) = \prod_{j=1}^{n} P(W_j | Z_1, ..., Z_m). \tag{3}$$

In Eq. (3), each $W_j$ is assumed to be conditionally independent from all the other textual variables given the visual variables. The empirical results reported in (Tschannen et al., 2023) do not show a clear winner between the AR and the parallel prediction, and the authors use a mixture of the two strategies in training their VLM (Sec. 2). Conversely, in our experiments (Sec. 4.1 and 4.2) we show that our proposed disentangled factorization (Eq. (1)) is a better trade-off between the conditional independence of Eq. (3) and the standard image captioning factorization of Eq. (2), and it also improves over the mixed strategy adopted in (Tschannen et al., 2023).

### 3.1 USING A DECODER FOR CAUSAL PREDICTION

In this section, we show how textual tokens can be generated using our CGM. Note that our goal is not image captioning, but we use our method, which we call Causally-Ordered Generative Training (COGT), for vision-language compositional understanding. Since CLIP is the most commonly adopted backbone by previous works on compositionality (Sec. 2), in the following we use CLIP as an example VLM, and in Sec. 4.2 we show additional results obtained with other VLMs.

We freeze the CLIP visual encoder ($\mathcal{E}$) and, from a given image $X$, we extract a set of features from the last ($L$) and the penultimate ($L-1$) layer of $\mathcal{E}$: $\mathcal{Z} = \{z^L_{[\text{CLS}]}, z^L_1 ..., z^L_p, z^{L-1}_{[\text{CLS}]}, z^{L-1}_1 ..., z^{L-1}_p\}$. For the $l$-th layer of the encoder, $z^l_{[\text{CLS}]}$ is the embedding vector of the class token (Dosovitskiy et al., 2021), while $z^l_1 ..., z^l_p$ are the embedding vectors of the patch tokens. Using a grid of $p$ patch tokens, we have $m = 2p + 2$. We use the embedding vectors of the penultimate layer jointly with the last layer features to help the model reasoning about smaller resolution objects. Indeed, previous work (Ghiasi et al., 2022; Wysoczańska et al., 2024) showed that there is usually a decrease in the amount of spatial information represented in the last layer of CLIP. Moreover, we use a mapping network $\mathcal{M}$ (Fig. 2) to reduce the dimensionality of the visual features to match the decoder embedding size. $\mathcal{M}$ is composed of a linear layer, preceded and followed by LayerNorm, and all features in $\mathcal{Z}$ are obtained as output of $\mathcal{M}$. The parameters of $\mathcal{M}$ are learned jointly with our decoder (see below) and $\mathcal{M}$ is shared by all features in $\mathcal{Z}$ and both layers of $\mathcal{E}$ ($L$ and $L-1$).

We replace the CLIP textual encoder with our decoder $\mathcal{D}$, a relatively small network, composed of only three blocks with ∼64M total parameters, which is the module we use to adapt CLIP to solve compositional tasks. Fig. 2 shows the architecture of $\mathcal{D}$, which takes as input $2n$ tokens. The first sequence of $n$ tokens are masked tokens, while the others are visible tokens, and we represent each $w_j$ with both a masked and a visible token. Specifically, to condition $\mathcal{D}$ with respect to the event $S_j = t$ ($t \in V$) in Eq. (1), we use masked tokens specific for each syntactic type $t$ in $V$. In more detail, $V$ is composed of the 45 standard syntactic categories defined in (Silveira et al., 2014) (see App. F). We associate each category $t$ with a masked token $\text{MSK}_t$. Then, for each word $w_j \in C$, if $type(w_j) = t$, then the masked token used for $w_j$ is $\text{MSK}_t$. This is simply implemented using a lookup table of masked token embeddings, composed of 45 different initial embedding vectors (learned using standard backpropagation) and which extends the (single) masked token used in common masked-token prediction tasks (Kenton & Toutanova, 2019). The other $n$ tokens are visible, standard textual tokens, one for each $w_j \in C$. In this way, $w_j$ is represented both as a visible token and as a masked token of type $t$. In a given layer of $\mathcal{D}$, these two tokens are respectively represented by the masked-token embedding vector $m_j$ and the visible-token embedding vector $v_j$.

Each block of $\mathcal{D}$ is composed of two layers. In the first layer, we compute the self-attention of each masked embedding $m_j$ with itself, jointly with the attention of $m_j$ with all the visible embeddings $v_{i_1}, ..., v_{i_k}$, where $\mathbf{PA}(W_j) = \{W_{i_1}, ..., W_{i_k}, S_j, Z_1, ..., Z_m\}$. Note that there is no attention between $m_{j_1}$ and $m_{j_2}$, with $j_1 \neq j_2$. In the same layer, we compute the self-attention of each visible embedding $v_j$ with itself, jointly with the attention of $v_j$ with $v_{i_1}, ..., v_{i_k}$ (Fig. 2). Note that there is no information leak, since $m_j$, later used for the final prediction, has no direct or indirect access to $v_j$. We call this *Dependency Guided Attention* to differentiate it from the standard self-attention (Fig. 2). In the second layer of each block of $\mathcal{D}$, both the masked ($m_j$) and the visible ($v_j$) embeddings pay attention to the visual features in $\mathcal{Z}$ using cross-attention, in this way implementing the dependence between $W_j$ and $Z_1, ..., Z_m$. Finally, after the last block of $\mathcal{D}$ we discard the visible-token embeddings and we fed each masked-token final embedding to a linear layer computing a posterior distribution over the vocabulary of textual terms.

$\mathcal{D}$ is trained from scratch using as the only objective the maximization of the log-likelihood of the disentangled factorization:

$$\mathcal{L} = \log \left[ \prod_{j=1}^{n} P(W_j | \mathbf{PA}(W_j)) \right] = \sum_{j=1}^{n} \log(P(W_j | \mathbf{PA}(W_j))). \tag{4}$$

**Inference** Most of the compositional tasks are modeled as image-to-text retrieval tasks. In case of COGT, given a testing image $X$, we compute the log-likelihood of all the candidate testing captions and we select the highest scoring sentence. Note that computing $\mathcal{Z}$ is independent of the specific

caption $C$, and it can be done once for each image in the dataset. The log-likelihood is computed using Eq. (4) and a semi-parallel AR prediction strategy which follows the partial order induced by the DT. Specifically, using the dependency parser we extract $\mathcal{T}$ from a candidate caption $C$. Then we proceed using a *level order traversal* of $\mathcal{T}$, in which, starting from the root, we predict in parallel all the tokens of a given level of the tree and then we move to the next level.

# 4 EXPERIMENTS

In our evaluation we use four common compositional benchmarks: ARO (Yuksekgonul et al., 2023), SugarCrepe (Hsieh et al., 2024), VL-CheckList (Zhao et al., 2022) and ColorSwap (Burapacheep et al., 2024), and an additional benchmark FG-OVD (Bianchi et al., 2024) which we propose in this paper. Most of them are composed of different tasks and datasets, and we report both the task-specific and the average accuracy across all tasks. Following (Zhang et al., 2024), we do not use COCO Order and Flickr Order (two of the ARO tasks) because it has been previously showed that a "blind" LM, *with no access to the image*, can achieve about $99\%$ accuracy on these tasks (Tschannen et al., 2023). The reason of this is due to the grammatical and semantic errors introduced in the negatives when swapping or replacing caption words (see Sec. 1). For instance, the sentence "with man is wearing ears the an glasses pierced orange hat and" (Flickr Order) can be easily detected as false by an LM without any visual knowledge. In App. C.2 we show additional results with Winoground (Thrush et al., 2022), which, however, is often not used by other methods based on CLIP (Zhang et al., 2024) because requires the VLM to be able to detect out-of-focus objects in low-resolution images (Diwan et al., 2022). In contrast, we propose to use FG-OVD (Bianchi et al., 2024), a benchmark originally proposed to evaluate the ability of open-vocabulary object detectors to discern fine-grained object properties. In FG-OVD, negative captions are created starting from the object-specific captions by replacing attributes referring to the object's color, material, texture, etc. We crop the objects' bounding boxes which we use jointly with positive and negative captions and an image-to-text retrieval task (more details in App. B).

Table 1: Comparison between different generative training strategies. The value **+x** reported in the $i$-th row, column Average, refers to the average improvement across all datasets with respect to the method in row $i - 1$.

| Model | ARO | | | SugarCrepe | | | | VL-Checklist | | | | ColorSwap | FG-OVD | Avg |
|---|---|---|---|---|---|---|---|---|---|---|---|---|---|---|
| | Relation | Attribute | Avg | Add | Replace | Swap | Avg | Attribute | Object | Relation | Avg | ITT | Avg | |
| Fully-Parallel | 76.37 | 49.24 | 62.81 | 98.98 | 77.98 | 68.81 | 81.92 | 83.66 | 67.55 | 74.7 | 75.3 | 25.24 | 41.84 | **57.42** |
| Mixed | 84.83 | 69.12 | 76.98 | 99.01 | 84.17 | 78.39 | 87.19 | 85.89 | 76.96 | 91.44 | 84.76 | 41.33 | 45.21 | **67.10**$_{+9.67}$ |
| Sequential-AR | 84.86 | 77.87 | 81.37 | 98.96 | 83.62 | 81.50 | 88.02 | 86.82 | 75.45 | 91.11 | 84.45 | 46.33 | 46.24 | **69.28**$_{+2.18}$ |
| COGT | 87.56 | 90.26 | 88.91 | 98.26 | 87.10 | 83.14 | 89.50 | 86.07 | 78.91 | 89.37 | 84.78 | 61.33 | 51.48 | **75.20**$_{+5.92}$ |

## 4.1 ABLATIONS

In the experiments of this section we follow a widely adopted protocol, first proposed in (Yuksekgonul et al., 2023), in which the VLM backbone is CLIP and the only training dataset is COCO (Lin et al., 2014). However, we *do not* use the hard negatives of (Yuksekgonul et al., 2023) for training because of their frequent semantic and syntactic errors (see above). In Tab. 1 we compare to each other the different word prediction strategies described in Sec. 3 using CLIP as the VLM. Specifically, we indicate with *Sequential-AR* the replacement of our decoder with a standard AR decoder (cross-attention over $\mathcal{Z}$ and standard causal attention over the past words of the caption), trained using the common image captioning objective defined in Eq. (2). Similarly to COGT, we freeze the CLIP encoder and we use the visual features ($\mathcal{Z}$) extracted from both the last and the penultimate layer of $\mathcal{E}$ (Sec. 3.1). We use a same-size decoder, which takes only visible words as input (details in App. D). *Sequential-AR* can be considered as our re-implementation of Cap[3] (Tschannen et al., 2023), trained on COCO and with a frozen visual encoder, which can directly be compared with the CGM-based strategy of COGT. Similarly, we indicate with *Fully-Parallel* our re-implementation of the parallel prediction strategy proposed in (Tschannen et al., 2023), using a decoder which takes as input only masked tokens (only cross-attention over $\mathcal{Z}$ with a frozen visual encoder), trained using Eq. (3). Finally, in *Mixed* we use the sequential-parallel mixed strategy adopted in CapPa, in

---

[3]There are no publicly available network weights for (Tschannen et al., 2023).

Table 2: Empirical contribution of different components of COGT.

| Parser | Mask-Specific | Layers | ARO Relation | ARO Attribute | ARO Avg | SugarCrepe Add | SugarCrepe Replace | SugarCrepe Swap | SugarCrepe Avg | VL-Checklist Attribute | VL-Checklist Object | VL-Checklist Relation | VL-Checklist Avg | ColorSwap ITT | FG-OVD Avg | Avg |
|---|---|---|---|---|---|---|---|---|---|---|---|---|---|---|---|---|
| CRFPar | ✓ | 2 | 85.68 | 88.34 | 87.01 | 98.16 | 84.94 | 80.30 | 87.80 | 86.99 | 77.68 | 87.09 | 83.92 | 56.33 | 43.74 | 71.76 |
| Deep Biaffine | ✓ | 2 | 86.56 | 89.10 | 87.83 | 98.11 | 85.80 | 81.49 | 88.46 | 87.02 | 78.30 | 87.75 | 84.35 | 61.33 | 44.74 | 73.34 |
| Deep Biaffine + RoBERTa | ✗ | 2 | 84.75 | 86.16 | 85.46 | 98.86 | 84.37 | 80.25 | 87.82 | 83.79 | 78.24 | 90.84 | 84.29 | 58.00 | 46.99 | 72.51 |
| Deep Biaffine + RoBERTa | ✓ | 1 | 86.82 | 89.67 | 88.25 | 98.26 | 86.56 | 82.33 | 89.05 | 84.41 | 78.94 | 89.54 | 84.30 | 45.00 | 45.63 | 70.45 |
| Deep Biaffine + RoBERTa | ✓ | 2 | 87.56 | 90.26 | 88.91 | 98.26 | 87.10 | 83.14 | 89.50 | 86.07 | 78.91 | 89.37 | 84.78 | 61.33 | 51.48 | 75.20 |

which, following (Tschannen et al., 2023), we use 75% of the training samples with a parallel prediction (Eq. (3)) and 25% of the samples with an AR prediction (Eq. (2)). Architectural details are provided in App. D. The results show that COGT outperforms all the other prediction strategies in all the datasets, often with a significant margin. For instance, COGT achieves an average accuracy improvement of $+17.77$ points across all datasets with respect to *Fully-Parallel*, which arguably shows that the conditional independence assumption in Eq. (3) is too strong. Overall, these results confirm that an off-the-shelf dependency parser provides a priori knowledge which can be exploited to model the conditional dependencies between words in a sentence.

We further investigate the role of the dependency parser in Tab. 2. Specifically, the column *Parser* refers to the adopted dependency parser, where we compare 3 different methods: Deep Biaffine (Dozat & Manning, 2016), CRFPar (Zhang et al., 2020) and Deep Biaffine + RoBERTa (Dozat & Manning, 2016). Note that we use the parsers as black boxes, without any training or fine-tuning, and the differences in the corresponding rows of Tab. 2 are based only in the use of a different external parser for COGT. Tab. 2 shows that the best results correspond to the use of Deep Biaffine + RoBERTa (Dozat & Manning, 2016), which is aligned with the higher accuracy of this parser compared to the other two according to the linguistic leaderboard Penn Tree Bank (Marcus et al., 1993). Note also that, according to this widely adopted parser ranking (Marcus et al., 1993), there are higher performing parsers (e.g., (Mrini et al., 2019)), however their code is not publicly available or it is not easy to use. Thus, we opted for Deep Biaffine + RoBERTa (used in all the other experiments of this paper). However, the results in Tab. 2 show that, using a better parser, COGT can most likely achieve even better results.

The *Mask-Specific* column in Tab. 2 indicates the use of a dedicated masked token for each of the 45 syntactic categories of $V$ (Sec. 3.1), which is compared with a generic BERT-like masked token (Kenton & Toutanova, 2019). In the latter case, we use the same masked token initial embedding vector for all the $n$ masked tokens fed to $\mathcal{D}$ (replicated $n$ times), thus dropping any conditioning on $S_j$ in Eq. (1). The results in Tab. 2 show that this corresponds to a $-2.69$ point drop in accuracy averaged across all five datasets.

Finally, the *Layers* column in Tab. 2 indicates the number of layers of CLIP we use to extract the visual features $\mathcal{Z}$: *Layers* = 1 means only the last layer ($L$); *Layers* = 2 means that we use also the penultimate layer (Sec. 3.1). When the last layer only is used, the average accuracy drop is $-4.75$, showing the importance of using lower-level features in compositionality tasks where the VLM needs to consider small, non-foreground objects.

In App. C.1 we provide the complete version of Tab. 2 with all the possible combinations between the values of *Parser*, *Mask-Specific* and *Layers*, which confirm the results shown here.

## 4.2 MAIN EXPERIMENTS

**Setting.** In this section we compare COGT with state-of-the-art compositional methods. Since different works are based on different VLMs and use different training data, to make the comparison as fair as possible, we split our evaluation based on both the VLM backbone and the used training set. Specifically, in Tab. 3 we group the approaches based on CLIP (Radford et al., 2021) and in Tab. 4 those which adopt a different VLM, while Tab. 5 is dedicated to methods based on VLMs pre-trained using a language based decoder. In the first category, our approach is indicated by COGT-CLIP. In the second group, we use XVLM (Zeng et al., 2022) as our backbone (COGT-XVLM). Finally, we use InstructBLIP (Dai et al., 2023) for the VLM category with a language-based decoder (COGT-InstructBLIP) (more details in App. D). COGT-CLIP, COGT-XVLM and COGT-InstructBLIP are trained on COCO *only* ($\sim$ 100K training samples, see Sec. 4.1). Moreover, following (Zhang et al., 2024), we present additional results training on a combination of three

datasets: COCO, CC3M (Sharma et al., 2018), and Visual Genome (Krishna et al., 2017), and we call the corresponding methods as COGT-CLIP+, COGT-XVLM+ and COGT-InstructBLIP+. In this case, we use a decoder $\mathcal{D}$ with four blocks. Note that we use only $\sim 50K$ samples from Visual Genome because we *removed those training data which overlap with ARO and VL-Checklist*. On the other hand, CC3M ($\sim$3.3M training samples) is a much larger but also noisier dataset, since its captions are obtained from the Alt-text HTML attribute associated with web images, and we use it also to indirectly evaluate the robustness of COGT to noisy textual descriptions (see App. C.1). For each compared baseline, the results shown in the tables refer to the values reported in the original article (when available) or to our reproduction using the (possibly available) public code. The results on FG-OVD are averaged over all tasks and we report in App. C.3 the task-specific values.

Table 3: Comparison with compositional methods based on CLIP. For each baseline, we report the values published in the original article. In case a given dataset was not used by that baseline, but a public code is available, we report the results obtained by our reproduction. With **x**, $\underline{x}$ and $x^*$ we indicate the first, the second and the third best result, respectively.

| Model | ARO | | | SugarCrepe | | | | VL-Checklist | | | | ColorSwap | FG-OVD | Avg |
|---|---|---|---|---|---|---|---|---|---|---|---|---|---|---|
| | Relation | Attribute | Avg | Add | Replace | Swap | Avg | Attribute | Object | Relation | Avg | ITT | Avg | |
| *Zero-shot* | | | | | | | | | | | | | | |
| CLIP (Radford et al., 2021) | 59.00 | 62.00 | 60.50 | 85.58 | 80.76 | 70.83 | 79.05 | 67.93 | 82.83 | 64.19 | 71.65 | 35.67* | 47.33 | 58.84 |
| *Training on COCO only* | | | | | | | | | | | | | | |
| CLIP Fine-Tuned (Yuksekgonul et al., 2023) | 63.00 | 65.00 | 64.00 | . | | | . | | | | . | . | . | . |
| NegCLIP (Yuksekgonul et al., 2023) | 81.00 | 71.00 | 76.00 | 87.29 | 85.36 | 75.30 | 82.65 | 72.24 | 87.00 | 71.39 | 76.87 | 35.67* | 41.69 | 62.57 |
| CE-CLIP (Zhang et al., 2024) | 83.00 | 76.40 | 79.70 | 92.90 | 87.00 | 74.90 | 84.94 | 72.60 | 84.60 | 71.80 | 76.30 | 18.67 | 41.97 | 60.31 |
| Structure-CLIP (Huang et al., 2024) | 85.10* | 83.50* | 84.30* | . | | | . | | | | . | . | . | . |
| GNM (Sahin et al., 2024) | 65.00 | 65.00 | 65.00 | 82.85 | 80.95 | 66.71 | 76.83 | 70.15 | 85.91 | 64.10 | 73.38 | 13.00 | 38.79 | 53.40 |
| Plausible Adj. Neg (Buettner & Kovashka, 2024) | 65.07 | 67.94 | 66.51 | 89.64 | 85.37 | 70.88 | 81.96 | 76.51 | 88.13 | 69.90 | 78.17 | 17.67 | 44.98 | 57.86 |
| SDS-CLIP (Basu et al., 2023) | 55.00 | 66.00 | 60.50 | . | | | . | | | | . | . | . | . |
| COGT-CLIP | 87.56 | 90.26 | 88.91 | 98.26 | 87.10* | 83.14 | 89.50 | 86.07 | 78.91 | 89.37* | 84.78 | 61.33 | 51.48 | 75.20 |
| *Training on datasets larger than COCO* | | | | | | | | | | | | | | |
| CE-CLIP+ (Zhang et al., 2024) | 83.60 | 77.10 | 80.35 | 94.40 | 89.30 | 78.00* | 87.23* | 76.70 | 86.30 | 74.70 | 79.23 | . | . | . |
| IL-CLIP (Zheng et al., 2024) | . | | . | 73.80 | 73.00 | 62.90 | 69.90 | . | | | . | . | . | . |
| syn-CyCLIP (Cascante-Bonilla et al., 2023) | 69.00 | 63.65 | 66.33 | . | | | . | 68.06 | . | 65.73 | . | . | . | . |
| CLIP-SPEC (Peng et al., 2024) | 73.70 | 66.40 | 70.05 | . | | | . | | | | . | . | . | . |
| DAC-SAM (Doveh et al., 2023a) | 77.16 | 70.50 | 73.83 | 92.87 | 86.18 | 71.06 | 83.37 | 75.80 | 88.50 | 89.80 | 84.70* | 16.33 | 48.36 | 61.31 |
| DAC-LLM (Doveh et al., 2023a) | 81.28 | 73.91 | 77.60 | 95.83* | 88.09 | 72.48 | 85.47 | 77.30* | 87.30* | 86.40 | 83.66 | 18.33 | 49.60* | 62.93* |
| COGT-CLIP+ | 90.67 | 96.01 | 93.34 | 98.42 | 87.05 | 84.21 | 89.89 | 90.71 | 84.91 | 92.33 | 89.31 | 81.66 | 69.96 | 84.83 |

**CLIP based methods.** Tab. 3 shows that COGT-CLIP *largely* outperforms all the other approaches trained only on COCO **and it also outperforms all the methods trained on datasets larger than COCO**. For instance, using the average across all the datasets, COGT-CLIP outperforms the second best result in Tab. 3 (DAC-LLM (Doveh et al., 2023a)) by a remarkable 12.27 points. Note that DAC-LLM was trained on CC3M, a dataset an order of magnitude larger than COCO, and using high-quality LLM-based annotations (Sec. 2). Moreover, even considering the average computed on the individual datasets, COGT-CLIP outperforms all the other methods in Tab. 3 (including those trained on datasets larger than COCO). We believe that these results show that our CGM-based training strategy can better generalize by leveraging available training data, most likely because we remove spurious inter-variable associations from the learning objective (Sec. 3). Moreover, COGT-CLIP+ achieves even better results, with an average across all benchmarks that is almost 22 points more than the second best result (DAC-LLM).

Table 4: Comparison with methods based on other VLMs. Similar to Tab. 3, the baseline results are either taken from the original paper or reproduced using the public code.

| Model | ARO | | | SugarCrepe | | | | VL-Checklist | | | | ColorSwap | FG-OVD | Avg |
|---|---|---|---|---|---|---|---|---|---|---|---|---|---|---|
| | Relation | Attribute | Avg | Add | Replace | Swap | Avg | Attribute | Object | Relation | Avg | ITT | Avg | |
| *Zero-shot* | | | | | | | | | | | | | | |
| XVLM (Zeng et al., 2022) | 73.40 | 86.80 | 80.10 | . | | | . | 75.10* | 85.80 | 70.40 | 76.50 | . | . | . |
| *Training on COCO only* | | | | | | | | | | | | | | |
| CE-XVLM (Zhang et al., 2024) | 73.90* | 89.30* | 81.60* | . | | | . | 74.80 | 86.90 | 79.70* | 78.60* | . | . | . |
| HardNeg-DiffusionITM (Krojer et al., 2023) | 52.30 | 67.60 | 59.95 | . | | | . | | | | . | . | . | . |
| COGT-XVLM | 87.64 | 92.30 | 89.97 | 98.65 | 89.17 | 84.37 | 90.73 | 85.87 | 80.49 | 88.74 | 85.03 | 69.67 | 50.12 | 77.10 |
| *Training on datasets larger than COCO* | | | | | | | | | | | | | | |
| COGT-XVLM+ | 91.71 | 96.59 | 94.15 | 98.30 | 88.97 | 86.49 | 91.25 | 91.54 | 84.73* | 92.33 | 89.53 | 82.33 | 74.22 | 86.30 |

**Other VLMs.** The results in Tab. 3 are confirmed by those reported in Tab. 4, where COGT-XVLM largely outperforms the other methods, included CE-XVLM (Zhang et al., 2024), which uses our same VLM (XVLM) and the same training data (COCO). COGT-XVLM+ further improves these results and it also outperforms COGT-CLIP+. This is probably because the XVLM encoder can better represent small-scale objects than the CLIP encoder, and these objects are often referenced in the captions of these compositional benchmarks (Yuksekgonul et al., 2023).

Table 5: Comparison with methods based on encoder-decoder VLM architectures pre-trained with a textual token prediction task. [†] Lin et al. (2024) show additional results with $\alpha$ set using dataset-specific cross-validation data, which we do not report, however, to make the comparison fair to other methods that do not have access to benchmark data.

| Model | ARO | | | SugarCrepe | | | | VL-Checklist | | | | ColorSwap | FG-OVD | Avg |
|---|---|---|---|---|---|---|---|---|---|---|---|---|---|---|
| | Relation | Attribute | Avg | Add | Replace | Swap | Avg | Attribute | Object | Relation | Avg | ITT | Avg | |
| BLIP (Li et al., 2022) | 59.00 | 88.00 | 73.50 | . | . | . | . | 75.20 | 82.20 | 70.50 | 75.70 | . | . | . |
| BLIP2 (Li et al., 2023c) | 41.20 | 71.30 | 56.25 | . | . | . | . | 77.80 | 84.90 | 70.60 | 77.80 | . | . | . |
| InstructBLIP (FlanT5XL) (Dai et al., 2023) | 69.20 | 50.83 | 60.02 | 65.43 | 72.59 | 63.41 | 67.14 | 56.37 | 80.33 | 53.34 | 63.35 | 40.33* | 26.80* | 51.53* |
| MiniGPT-4 (Zhu et al., 2024) | 46.90 | 55.70 | 51.30 | . | . | . | . | 71.30 | 84.20 | . | . | . | . | . |
| GPT-4V (OpenAI, 2023) | . | . | . | 91.68 | 93.37 | 86.61 | 90.55 | . | . | . | . | . | . | . |
| LLaVA-1.5-13B (Liu et al., 2023) | . | . | . | . | . | . | 80.95 | . | . | . | . | . | . | . |
| LLaVA-1.5-13B+CRG (Wan et al., 2024) | . | . | . | . | . | . | 87.90 | . | . | . | . | . | . | . |
| LLaVA-1.6-34B (Liu et al., 2024) | . | . | . | . | . | . | 81.25 | . | . | . | . | . | . | . |
| LLaVA-1.6-34B+CRG (Wan et al., 2024) | . | . | . | . | . | . | 90.75 | . | . | . | . | . | . | . |
| BLIP-VisualGPTScore ($\alpha = 0$) (Lin et al., 2024) † | 89.10* | 95.30 | 92.20* | 91.00 | 93.30 | 91.00 | 91.77* | 78.70* | 92.60 | 90.80 | 87.37 | . | . | . |
| BLIP2-VisualGPTScore ($\alpha = 0$) (Lin et al., 2024) † | 90.70 | 94.30* | 92.50 | 92.70 | 93.00* | 91.24 | 92.31 | 73.90 | 90.10 | 89.90* | 84.63 | . | . | . |
| Cap (Tschannen et al., 2023) | 86.60 | 88.90 | 87.75 | 98.94 | 88.21 | 84.00 | 90.38 | . | . | . | . | . | . | . |
| CapPa (Tschannen et al., 2023) | 86.70 | 85.70 | 86.20 | 99.13 | 87.67 | 83.11 | 89.97 | . | . | . | . | . | . | . |
| COGT-InstructBLIP | 87.63 | 88.93 | 88.28 | 98.55† | 90.61 | 88.12 | 92.42 | 85.77 | 79.96 | 89.14 | 84.96 | 72.66 | 51.26 | 77.87 |
| COGT-InstructBLIP+ | 91.12 | 95.64 | 93.38 | 98.45 | 90.27 | 88.22† | 92.31 | 90.80 | 85.17* | 92.80 | 89.60 | 83.33 | 70.72 | 85.87 |

Table 6: Comparison of CLIP-based models using image classification tasks and linear probing.

| Model | CIFAR10 | CIFAR100 | ImageNet1K (top 1) | ImageNet1K (top 5) |
|---|---|---|---|---|
| CLIP (Radford et al., 2021) | 94.2 | 79.0 | 75.0 | 93.2 |
| CLIP Fine-Tuned (Yuksekgonul et al., 2023) | 95.0 | 80.0 | 74.0 | - |
| NegCLIP (Yuksekgonul et al., 2023) | 94.0 | 79.0 | 72.0 | - |
| CE-CLIP (Zhang et al., 2024) | 93.8 | 78.0 | - | 92.6 |
| CE-CLIP+ (Zhang et al., 2024) | 93.8 | 78.1 | - | 92.7 |
| COGT-CLIP | 96.7 | 84.9 | 74.4 | 93.3 |
| COGT-CLIP+ | 96.8 | 85.4 | 75.3 | 93.8 |

**Language-decoding based VLMs.** In Tab. 5 we compare to each other VLMs pre-trained using a decoder and a generative word prediction task. The compositional skills of these methods are generally much higher than the other VLMs, which indirectly confirms that a word-prediction training helps the VLM to understand the compositional nature of the human language (Sec. 2). However, a direct comparison with VLMs such as CLIP, XVLM or Stable Diffusion (Krojer et al., 2023) is difficult, since each of these backbones has been pre-trained on datasets with a huge difference in size. For instance, Cap and CapPa (Tschannen et al., 2023) were pre-trained with a private dataset composed of 1B image/Alt-text pairs (Tschannen et al., 2023), which is different orders of magnitude larger than the dataset used to pre-train XVLM ($\sim$ 16M training samples (Zeng et al., 2022)). Despite that, COGT-XVLM+ (Tab. 4) outperforms both Cap and CapPa and all the other methods in Tab. 5. Similarly, COGT-InstructBLIP+ significantly outperforms the zero-shot accuracy of InstructBLIP and, jointly with COGT-XVLM+, establishes *a new state of the art on each of these compositional datasets*.

## 4.3 DOWNSTREAM TASKS

Doveh et al. (2023b) show that most compositional methods usually deteriorate the VLM skills on non-compositional, standard tasks. We analyze this aspect using the protocol adopted by (Yuksekgonul et al., 2023; Zhang et al., 2024), which is based on linear probing the fine-tuned CLIP visual encoder on CIFAR10, CIFAR100 and ImageNet. Since in COGT $\mathcal{E}$ is frozen, we use $\mathcal{E}$ jointly with our mapping network $\mathcal{M}$ and, specifically, the feature $z_{[\text{CLS}]}^L$. The results shown in Tab. 6 show that COGT not only does it not deteriorate CLIP's features but it can even improve them.

## 5 CONCLUSION

In this paper we presented COGT, a compositional method based on a semi-parallel generative training. Specifically, we exploit the a priori knowledge of an off-the-shelf dependency parser to define a set of causal relations between the words of a sentence. These relations, collectively represented using a CGM, are used to make sparser the joint probability distribution of the textual variables by removing possible spurious inter-variable associations. As a result, COGT can better exploit the training data and reduce the risk of overfitting. Using extensive experiments, we showed that COGT is much more effective than standard AR or fully-parallel generative predictions and it largely outperforms all previous compositional works, including methods trained with much larger datasets.

ACKNOWLEDGMENTS

This work was supported by the PNRR project "Italian Strengthening of Esfri RI Resilience (IT-SERR)", funded by the European Union – NextGenerationEU (CUP B53C22001770006), by the EU Horizon project ELIAS (No. 101120237), and by the PNRR project "A Static and Dynamic Database for Historic Urban Contexts Accessibility", funded by the European Union – NextGenerationEU (CUP E53D23010500001).

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

## A  CAUSAL GRAFICAL MODELS

A Causal Graphical Model (CGM) over $n$ random variables $\mathbf{X} = \{X_1, ..., X_n\}$ is defined (Perry et al., 2022) as $\mathcal{M}(G, \mathbb{P_X})$, where: (1) $G$ is a directed acyclic graph with vertices $\mathbf{X}$ and edges $X_i \to X_j$ iff $X_i$ is a direct cause of $X_j$; (2) $\mathbb{P_X}$ is the joint distribution of $\mathbf{X}$ which follows the *disentangled (or causal) factorization* (Schölkopf et al., 2021; Perry et al., 2022):

$$P(X_1, ..., X_n) = \prod_{j=1}^{n} P(X_j | \mathbf{PA}(X_j)), \tag{5}$$

where $\mathbf{PA}(X_j)$ is the set of *parents* (direct causes) of $X_j$ in $G$. The difference between a CGM and a Directed Graphical Model is that the former assumes that $\mathbf{PA}(X_j)$ are direct causes of $X_j$. Although a formal proof that a statistical dependence is also a causal relation using only observational data is notoriously difficult (Pearl, 2009), in this paper we assume that the linguistic dependencies (Nivre, 2005) extracted by a dependency parser have a causal nature because they represent a strict linguistic association between the "head" and its "dependent". Specifically, Dependency Grammars (Sec. 2) can be considered as (probabilistic) generative grammars (Nivre, 2005; Chen & Manning, 2014; Obrêbski & Gralinski, 2004; Diaconescu, 2002), in which a dependence between a "head" word and its "dependent" word can be extracted using context-free generative rules. We interpret these rules as causal *mechanisms* (Schölkopf et al., 2021), which describe the causal influence of generating a specific "dependent" word given the value of the 'head' word. We leave as future work the possibility of replacing the dependency grammars used in this paper with other grammars such as, for instance, the causal grammars proposed in (Tenenbaum et al., 2007), as well as the possible introduction of counterfactual reasoning (Pearl, 2009) in our framework.

In Sec. 3, Eq. (1) is obtained using Eq. (5), the definition of conditional distribution and the assumption that $S_1, ..., S_n$ and $Z_1, ..., Z_m$ are independent of each other. Finally, the cardinality of $\{W_1, ..., W_{j-1}\}$ in Eq. (2) is, on average, $\frac{n}{2}$, while, assuming a balanced DT $\mathcal{T}$, the cardinality of $\{W_{i_1}, ..., W_{i_k}\} \subset \mathbf{PA}(W_j)$ is, on average, $O(\log(n))$. The consequence of this is that the conditional distributions learned at training time (Eq. (1)) and used at inference time (Eq. (4)) are sparser than those learned by a standard AR model (Sec. 3).

## B  THE FG-OVD DATASET

In this section, we describe the compositional benchmark based on the FG-OVD dataset which we propose in this paper and which was used in Sec. 4. The FG-OVD dataset was originally proposed to evaluate the fine-grained discriminative capabilities of open-vocabulary detectors in object detection tasks. Each image usually contains multiple objects, where each object is associated with both a bounding box and a corresponding caption. We use the bounding box to crop the image and the corresponding caption as the true caption. The cropped images are resized to a resolution of $224 \times 224$. Then, each object image is associated with several false captions (on average, ten), selected based on the original FG-OVD Trivial, Easy, Medium, and Hard tasks (Bianchi et al., 2024). Specifically, in the Trivial task, negative captions are randomly sampled from unrelated objects (of different images), offering a basic challenge for retrieval. The Easy, Medium, and Hard tasks progressively increase the difficulty by generating negative captions starting from the true caption. For instance, in the Easy task, three attributes of the true caption are replaced with three unrelated attributes. In the Medium and the Hard task, two and one attributes are replaced, respectively. The rationale is that the less the true sentence is modified, the harder is to distinguish the false from the true captions (Bianchi et al., 2024). Specifically, as fewer attributes are replaced, the distinction between correct and incorrect captions becomes more subtle, requiring the model to capture increasingly fine-grained, compositional details in the image-text pairs. Tab. 11 reports the number of testing images for each task, while different examples are shown in App. E.1.

Table 7: Extended version of Tab. 2

| Parser | Layers | Mask Specific | ARO | | | SugarCrepe | | | | VL-Checklist | | | | ColorSwap | FG-OVD | Avg |
|---|---|---|---|---|---|---|---|---|---|---|---|---|---|---|---|---|
| | | | Relation | Attribute | Avg | Add | Replace | Swap | Avg | Attribute | Object | Relation | Avg | ITT | Avg | |
| Deep Biaffine | 1 | ✗ | 87.19 | 84.92 | 86.06 | 98.31 | 85.52 | 78.56 | 87.46 | 87.28 | 78.56 | 90.24 | 85.36 | 14.00 | 43.91 | 63.36 |
| Deep Biaffine | 1 | ✓ | 87.34 | 86.93 | 87.14 | 98.02 | 85.17 | 80.77 | 87.99 | 86.47 | 79.29 | 89.41 | 85.05 | 38.00 | 44.77 | 68.59 |
| Deep Biaffine | 2 | ✗ | 86.10 | 86.51 | 86.31 | 98.48 | 85.37 | 80.59 | 88.14 | 87.60 | 77.97 | 91.51 | 85.69 | 60.33 | 46.46 | 73.39 |
| Deep Biaffine | 2 | ✓ | 86.56 | 89.10 | 87.83 | 98.11 | 85.80 | 81.49 | 88.46 | 87.02 | 78.30 | 87.75 | 84.35 | 61.33 | 44.74 | 73.34 |
| CRFPar | 1 | ✗ | 86.46 | 84.36 | 87.83 | 98.45 | 84.15 | 78.65 | 87.08 | 87.51 | 77.04 | 90.15 | 84.90 | 14.00 | 45.83 | 63.93 |
| CRFPar | 1 | ✓ | 86.85 | 88.33 | 87.59 | 98.10 | 86.00 | 81.34 | 88.47 | 86.80 | 78.49 | 88.80 | 84.70 | 42.33 | 42.27 | 69.07 |
| CRFPar | 2 | ✗ | 85.53 | 86.67 | 86.10 | 98.74 | 85.58 | 81.84 | 88.72 | 87.43 | 77.33 | 90.30 | 85.02 | 59.33 | 44.08 | 72.65 |
| CRFPar | 2 | ✓ | 85.68 | 88.34 | 87.01 | 98.16 | 84.94 | 80.30 | 87.80 | 86.99 | 77.68 | 87.09 | 83.92 | 56.33 | 43.74 | 71.76 |
| Deep Biaffine + RoBERTa | 1 | ✗ | 86.82 | 86.76 | 86.79 | 98.69 | 86.59 | 80.35 | 88.54 | 85.31 | 79.11 | 89.88 | 84.76 | 28.33 | 43.92 | 66.47 |
| Deep Biaffine + RoBERTa | 1 | ✓ | 86.82 | 89.67 | 88.25 | 98.26 | 86.56 | 82.33 | 89.05 | 84.41 | 78.94 | 89.54 | 84.30 | 45.00 | 45.63 | 70.45 |
| Deep Biaffine + RoBERTa | 2 | ✗ | 84.75 | 86.16 | 85.46 | 98.86 | 84.37 | 80.25 | 87.82 | 83.79 | 78.24 | 90.84 | 84.29 | 58.00 | 46.99 | 72.51 |
| Deep Biaffine + RoBERTa | 2 | ✓ | 87.56 | 90.26 | 88.91 | 98.26 | 87.10 | 83.14 | 89.50 | 86.07 | 78.91 | 89.37 | 84.78 | 61.33 | 51.48 | 75.20 |

## C  ADDITIONAL EXPERIMENTS

### C.1  ADDITIONAL ABLATIONS

In Tab. 7 we show an extension of Tab. 2 containing all the possible combinations of the components analyzed in Tab. 2. For instance, this table shows the importance of using two visual feature layers in some of the datasets. Indeed, when both layers are used, the model can leverage not only high-level, abstract visual features but also more detailed, lower-level information. This is particularly important in tasks which require attention to object details, where the additional layer helps to capture a more nuanced representation of the input. For instance, in datasets like ColorSwap, this deeper feature extraction leads to drastic improvements (observed across all parsers), which is probably due to the better representation of the color/texture appearance in the lower level features. On the other hand, the use of mask-specific tokens also plays a significant role. Although the improvement magnitude varies depending on the dataset and the task, the overall trend indicates that using mask-specific tokens contributes positively to the accuracy.

Comparing the results of COGT-CLIP with COGT-CLIP+ and COGT-XVLM with COGT-XVLM+ (Tab. 3 and Tab. 4), the mean improvement of the larger-training versions with respect to the COCO-only trained models is about 9 points averaged across all the datasets. This shows that COGT can benefit from larger training and, indirectly, that the noisier captions in CC3M can be effectively parsed by our parser. Finally, a recent trend in VLM fine-tuning and/or pre-training adopts LLMs to create or re-write the textual annotations (Li et al., 2024c; Doveh et al., 2023a), which can in principle help the dependency parser with very noisy captions. We leave this as future work.

### C.2  WINOGROUND AND THE MULTIMODAL VISUAL PATTERN (MMVP) BENCHMARKS

In Tab. 8 we show additional results obtained using the Winoground (Thrush et al., 2022) and the MMVP (Tong et al., 2024) benchmarks. MMVP is a relatively small dataset, with 9 tasks but only 15 samples per task (135 total samples), thus performance measured on this benchmark has a limited statistical significance. Nevertheless, also in this dataset COGT-InstructBLIP+ largely outperforms the second best tested model (DAC-LLM) by 4.45 points. On the other hand, on Winoground, COGT-InstructBLIP+, despite an improvement of +7.5 points with respect to InstructBLIP zero-shot, it is largely outperformed by InstructBLIP-VQA Score (Lin et al., 2025). InstructBLIP-VQA could eventually be merged with COGT, but we leave this as a future work.

### C.3  FG-OVD

Tab. 10 reports the FG-OVD task-specific accuracy of the methods compared in Tab. 3. In the Zero-shot category, CLIP  (Radford et al., 2021) shows a solid performance on all but the Hard task (21.35 points accuracy). The same applies to those methods based on CLIP, such as NegCLIP (Yuksekgonul et al., 2023), GNM  (Sahin et al., 2024), Plausible Adj. Neg  (Buettner & Kovashka, 2024), and CE-CLIP  (Zhang et al., 2024), which show a noticeable drop in performance as the difficulty increases. Among these models, Plausible Adj. Neg  (Buettner & Kovashka, 2024) stands out with a relatively strong performance, especially on the Medium (43.13) and Easy (45.88) tasks. On the other hand, COGT-CLIP and COGT-XVLM demonstrate a significant improvement in per-

Table 8: Model comparison on MMVP and Winoground.

| | MMVP | Winoground |
|---|---|---|
| **Model** | **Group Score** | **Group Score** |
| CLIP (Radford et al., 2021) | 15.55 | 8.00 |
| NegCLIP (Yuksekgonul et al., 2023) | 26.67 | 8.00 |
| CE-CLIP (Zhang et al., 2024) | 19.25 | 5.00 |
| DAC-SAM (Doveh et al., 2023a) | 23.70 | 6.25 |
| DAC-LLM (Doveh et al., 2023a) | 28.14* | 3.25 |
| Plausible Adj. Neg (Buettner & Kovashka, 2024) | 21.48 | 6.75 |
| InstructBLIP (Dai et al., 2023) | 14.81 | 4.75 |
| InstructBLIP-VQA Score (Lin et al., 2025) | - | **28.50** |
| COGT-CLIP | 20.74 | 6.65 |
| COGT-CLIP+ | 26.67 | 9.75 |
| COGT-XVLM | 17.77 | 7.25 |
| COGT-XVLM+ | 28.88 | 10.50* |
| COGT+InstructBLIP | 28.88 | 7.75 |
| COGT+InstructBLIP+ | **32.59** | 12.25 |

formance across all difficulty levels, particularly on the Hard task, where COGT-CLIP achieves a score of 33.82, which is greater than all other methods, including models trained on larger datasets, such as DAC-SAM and DAC-LLM (Doveh et al., 2023a). These results highlight the strength of COGT when dealing with fine-grained compositional tasks. Finally, similarly to the results shown in Sec. 4.2, COGT-CLIP+, COGT-XVLM+ and COGT-InstrctBLIP+ further increase the advantage over other approaches.

### C.4 COMPUTATIONAL EFFICIENCY

Tab. 9 shows the training and inference times for COGT and CLIP-based models. COGT-CLIP and COGT-CLIP+ require respectively 8 and 72 hours to train on a *single* RTX A5000 GPU with a batch size of 128 using the datasets of Sec. 4.2. For comparison, we use DAC-SAM and DAC-LLM (Doveh et al., 2023a), which are the only models we know with publicly available training times that can be directly compared to COGT-CLIP+, as they are trained on a similar dataset of ~3.3M samples. In particular, both DAC-SAM and DAC-LLM complete their training in 12 hours on *six* V100 GPUs with a batch size of 32. However, this training time does not include the computationally intensive dense annotation generation pipeline (Sec. 2), which involves BLIP2 and SAM or GPT-Neo-2.7B (for DAC-SAM and DAC-LLM, rispectively). Similarly, the training times for COGT-CLIP and COGT-CLIP+ do not include the DT generations, which however involves a relatively quick preprocessing step (one DT per caption), taking approximately 3 minutes for COCO and 1.5 hours for the combined CC3M, COCO, and Visual Genome datasets. Moreover, we evaluate the computational costs of COGT and CLIP in terms of memory usage and inference time. COGT-CLIP, COGT-CLIP+, and CLIP require 0.73 GB, 0.88 GB, and 1.16 GB of memory, respectively, where the difference with respect to CLIP is mainly due to the fact that COGT does not use the CLIP textual encoder. Finally, for both COGT-CLIP and COGT-CLIP+, the inference times reported in Tab. 9 include an additional 0.01 seconds required for generating the DT of the testing caption using the Deep Biaffine + RoBERTa parser, and COGT-CLIP+ is slower than COGT-CLIP because of its larger (four blocks) decoder. All times are computed using an RTX A5000 GPU with a batch size of 32.

## D IMPLEMENTATION DETAILS

### D.1 ARCHITECTURES

In COGT-CLIP and in COGT-XVLM we use ViT-B/32 CLIP (Ilharco et al., 2021) and the Swin-Transformer of XVLM (Zeng et al., 2022) as the visual encoder, respectively. In both cases, we use both the last and the penultimate layer features of the encoder (Sec. 3.1). In COGT-InstructBLIP, we use the output of the InstructBLIP Q-Former (Dai et al., 2023) as the visual encoder. Since InstructBLIP needs a textual description of the task (called "instruction" (Dai et al., 2023)), COGT-InstructBLIP is trained using the prompts suggested in (Dai et al., 2023) for captioning tasks. At inference time, both the zero-shot results of InstructBLIP and those of COGT-InstructBLIP are

Table 9: A training and inference times comparison.

| Training | | | |
|---|---|---|---|
| **Model** | **Training Time (hrs)** | **Batch Size** | **GPU Setup** |
| COGT-CLIP | 8 | 128 | RTX A5000 (Single GPU) |
| COGT-CLIP+ | 72 | 128 | RTX A5000 (Single GPU) |
| DAC-SAM | 12 | 32 | V100 (Six GPUs) |
| DAC-LLM | 12 | 32 | V100 (Six GPUs) |
| **Inference** | | | |
| **Model** | **Inference Time (s)** | **Batch Size** | **GPU Setup** |
| COGT-CLIP | 0.07 | 32 | RTX A5000 (Single GPU) |
| COGT-CLIP+ | 0.09 | 32 | RTX A5000 (Single GPU) |
| CLIP | 0.06 | 32 | RTX A5000 (Single GPU) |

Table 10: FG-OVD: task specific results.

| Model | Hard | Medium | Easy | Trivial | Avg |
|---|---|---|---|---|---|
| *Zero-shot* | | | | | |
| CLIP (Radford et al., 2021) | 21.35 | 48.75 | 51.73 | 67.48[*] | 47.33 |
| InstructBLIP (FlanT5XL) (Dai et al., 2023) | 22.23 | 33.25 | 31.87 | 19.85 | 26.80 |
| *Training on COCO only* | | | | | |
| NegCLIP (Yuksekgonul et al., 2023) | 18.39 | 36.96 | 41.95 | 69.49 | 41.69 |
| GNM (Sahin et al., 2024) | 16.08 | 34.74 | 39.88 | 64.48 | 38.79 |
| Plausible Adj. Neg (Buettner & Kovashka, 2024) | 21.35 | 43.13 | 45.88 | **69.59** | 44.98 |
| CE-CLIP (Zhang et al., 2024) | 21.86 | 40.36 | 43.11 | 62.53 | 41.97 |
| Fully-Parallel | 25.22 | 47.41 | 54.04 | 40.72 | 41.84 |
| Mixed | 30.16 | 51.38 | 56.2 | 43.09 | 45.21 |
| Sequential-AR | 30.18 | 54.01 | 57.04 | 43.73 | 46.24 |
| COGT-CLIP | 33.82 | 59.30 | 61.35 | 51.43 | 51.48 |
| COGT-XVLM | 32.69 | 58.52 | 60.05 | 49.22 | 50.12 |
| COGT-InstructBLIP | 33.90 | 59.91 | 61.12 | 50.15 | 51.26 |
| *Training on datasets larger than COCO* | | | | | |
| DAC-SAM (Doveh et al., 2023a) | 26.00 | 48.65 | 53.73 | 65.05 | 48.36 |
| DAC-LLM (Doveh et al., 2023a) | 25.29 | 52.36 | 56.89 | 63.89 | 49.60 |
| COGT-CLIP+ | 55.40 | 81.50 | 85.29 | 57.65 | 69.96[*] |
| COGT-XVLM+ | **58.78** | **83.86** | **87.45** | 66.82 | **74.22** |
| COGT-InstructBLIP+ | 53.59[*] | 81.37[*] | 84.52[*] | 63.36 | 70.72 |

obtained using the prompt "Write a description for the photo.". The above considerations apply also to the COGT-X+ models.

Independently of the VLM encoder, the features $\mathcal{Z}$ are obtained using a mapping network $\mathcal{M}$ on top of the corresponding frozen visual encoder (Sec. 3.1). Our decoder consists of 3 blocks (respectively, 4 blocks in case of COGT-X+), each composed of a multi-head Dependency Guided Attention (Sec. 3.1) and a cross-attention layer. Each attention layer is composed of 8 attention heads, with embedding size equal to 512, while we use 12 heads and embedding size equal to 768 in the COGT-X+ models. We apply a dropout rate of 0.1 to the residual connections, the attention weights, and the embeddings.

The differences in the number of trainable parameters among the different baselines in Tab. 1 are only due to the size of the embedding dictionary. *Fully-Parallel* has an embedding dictionary consisting of only one MSK token, resulting in a total of 13 million trainable parameters, of which only 512 are dedicated to represent the MSK token. The total number of trainable parameters for *Mixed*, *Sequential-AR*, and COGT is approximately 64 million ($\mathcal{M}$ included). Among these, *Sequential-AR* uses an embedding dictionary that matches the size of the CLIP ViT-B/32 textual encoder, while *Mixed* introduces an additional MSK token for parallel processing. In contrast, COGT employs 45 extra MSK tokens, each representing a specific dependency relation extracted by the parser, resulting in a negligible increase in the total parameter count of only 0.04%. All the decoders in Tab. 1 are composed of three blocks which differ only in their attention masks, and they all alternate a textual-token embedding attention layer with a cross-attention layer with the features $\mathcal{Z}$ (extracted from the last and the penultimate layer of the CLIP visual encoder, see Sec. 3.1).

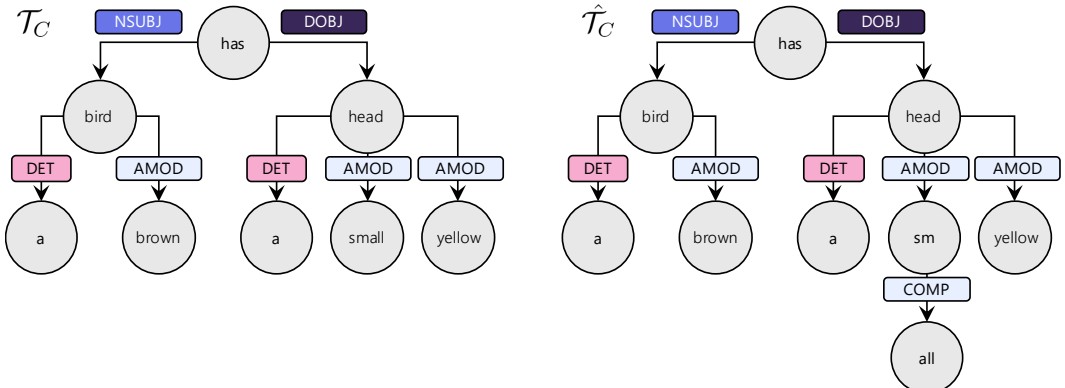

Figure 3: Dependency tree modifications due to the sub-word tokenization. On the left, the original dependency tree $\mathcal{T}$ represents the sentence "A brown bird with a small yellow head", and it is the output of a word-level dependency parser, where "small" is a single node. On the right, the modified tree, $\hat{\mathcal{T}}$, which accounts for sub-word tokenization by splitting the word 'small' into two nodes: "sm" and "all". A new syntactic relation, called comp, is introduced between these sub-word nodes.

### D.2 TOKENIZATION

The output of the dependency parser is a tree in which each node corresponds to a caption word. In contrast, the COGT decoder uses a standard sub-word tokenization, splitting words into smaller tokens which brings to a smaller embedding dictionary. This discrepancy leads to cases where a word of the dependency tree is split into multiple tokens by the COGT decoder's tokenizer. We modify the dependency tree to handle this mismatch: if a word $w_j$ is split into sub-tokens $w_{j_1}$ and $w_{j_2}$ by the COGT decoder's tokenizer, then we create a new node for $w_{j_2}$ and a new edge between $w_{j_2}$ and $w_{j_1}$ associated with a dedicated relation called comp (Fig. 3). Note that $w_j$ has been removed. As a result, in $\mathcal{G}$ we have that: $\mathbf{PA}(W_{j_1}) = \mathbf{PA}(W_j)$ and $\mathbf{PA}(W_{j_2}) = \mathbf{PA}(W_j) \cup \{W_{j_1}\}$.

### D.3 EXPERIMENTS

We use the architectures described above across all the experiments (e.g., the same number of blocks, learnable parameters, etc.). Specifically, we freeze the weights of the visual encoder and we train only our textual decoder. We train in mixed precision (FP16) with batch size set to 128 on a GPU RTX A5000 with 24GB of VRAM for 10 epochs. Following Yuksekgonul et al. (2023), we select the best checkpoint using the validation set provided in (Yuksekgonul et al., 2023). In all the datasets and in all the experiments, we use the Adam optimizer with an initial learning rate set to $5 \times 10^{-4}$. Finally, we apply a Cosine Annealing Learning Rate Scheduler with 50 warmup steps.

## E DATASETS AND TASKS

We provide details about the FG-OVD dataset and in App. B and we briefly summarize here the main characteristics of the other benchmarks. Tab. 11 shows the main statistics of each dataset and in App. E.1 we show a few images illustrating the benchmark typically tasks.

**ARO** (Yuksekgonul et al., 2023) is a VLM benchmark for compositional reasoning and word-order sensitivity. It is composed of two main tasks: Visual Genome Relation and Visual Genome Attribution. In the Visual Genome Relation task, the goal is to evaluate the models' ability to correctly interpret the relationships between objects. On the other hand, Visual Genome Attribution focuses on evaluating the ability to associate the correct attribute with the correct object. As mentioned in Sec. 4, we do not use COCO Order and Flickr Order because different authors recently found that grammatical errors in the generated captions of these datasets lead to tasks which can be solved purely relying on an LLM language prior (Zhang et al., 2024; Tschannen et al., 2023; Lin et al., 2024).

Table 11: Main statistics of the benchmarks.

| Dataset | Split | Number of testing samples | Avg. caption length (n. of words) |
|---|---|---|---|
| ARO | Relation | 23,937 | 8.1 |
| ARO | Attribution | 28,748 | 7.1 |
| SugarCrepe | Add | 2,754 | 12.9 |
| SugarCrepe | Replace | 3,846 | 11.5 |
| SugarCrepe | Swap | 911 | 13.5 |
| VL-Checklist | Attribute | 118,253 | 2.4 |
| VL-Checklist | Object | 389,357 | 3.2 |
| VL-Checklist | Relation | 75,641 | 3.5 |
| ColorSwap | - | 300 | 8.8 |
| FG-OVD | Hard | 3,545 | 10.4 |
| FG-OVD | Medium | 2,968 | 11.3 |
| FG-OVD | Easy | 1,299 | 16.2 |
| FG-OVD | Trivial | 3,545 | 9.7 |

**SugarCrepe** (Hsieh et al., 2024) is a dataset developed to evaluate how well VLMs can understand and process complex compositional tasks by presenting them with carefully designed hard negative examples. Drawing inspiration from datasets like CREPE (Ma et al., 2023), VL-CheckList (Zhao et al., 2022), and ARO (Yuksekgonul et al., 2023), SugarCrepe focuses on atomic concepts and their compositions, such as objects, attributes, and relations. The dataset is split into three tasks: "Replace", "Swap" and "Add". In "Replace", an atomic concept in the original text is replaced with a new, mismatched concept. A replacement can involve an object, an attribute or a relation. In "Swap", the negative caption is created by exchanging two atomic concepts of the same category without introducing new elements. In "Add", a new concept is added to the original caption, leading to a misalignment with the visual scene content.

**VL-Checklist** (Zhao et al., 2022) is a benchmark composed of four datasets: Visual Genome (Krishna et al., 2017), SWiG (Pratt et al., 2020), VAW (Pham et al., 2021), and HAKE (Li et al., 2019). Each image is associated with two descriptions: a true and a false caption. The true descriptions originate from the original image-text pairs in the datasets, while the false ones are generated by modifying a single word in the true description, altering its overall meaning. These false descriptions are organized into three main types: objects, attributes, and relations.

**ColorSwap** (Burapacheep et al., 2024) evaluates the ability of multimodal models to accurately associate objects with their corresponding colors. Each sample contains a caption-image pair along with a "color-swapped" pair. The two captions in each sample use the same text, but the relation between colors and objects are inverted.

### E.1 QUALITATIVE RESULTS

In Fig. 4-Fig. 12 we show some qualitative results in which we compare COGT-CLIP+ with the second best approach in Tab. 3 (DAC-LLM). We use these figures also to illustrate the tasks of the different benchmarks, with a special emphasis on FG-OVD, proposed in this paper.

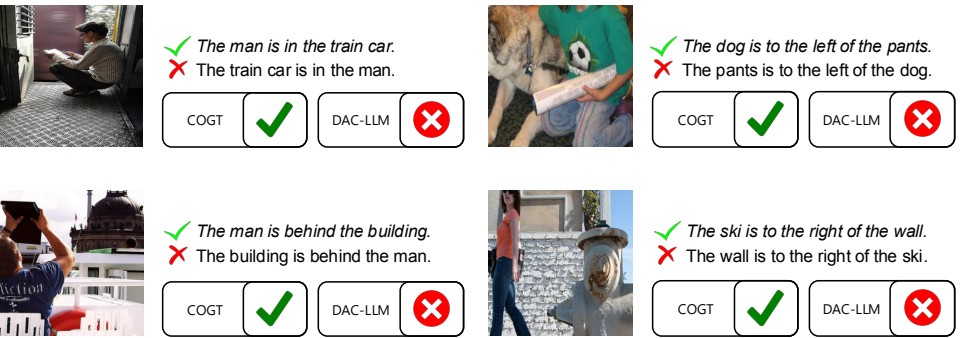

Figure 4: Qualitative results on sample images of the ARO Relation test split. We compare our approach with DAC-LLM which is the second best approach according to the results reported in Tab. 3.

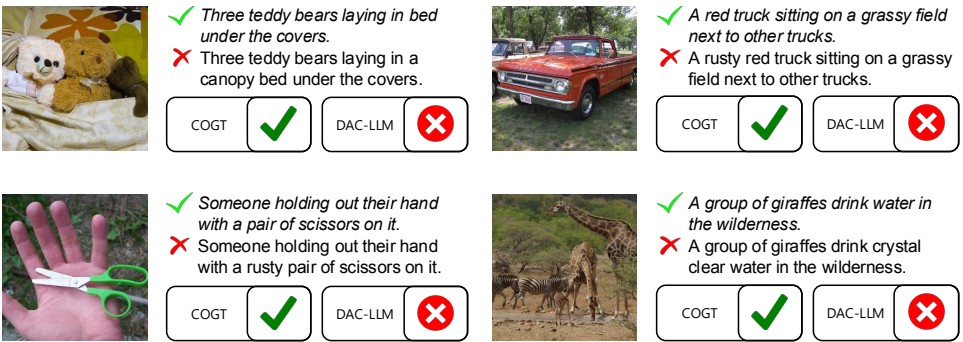

Figure 5: Qualitative results on sample images of SugarCrepe.

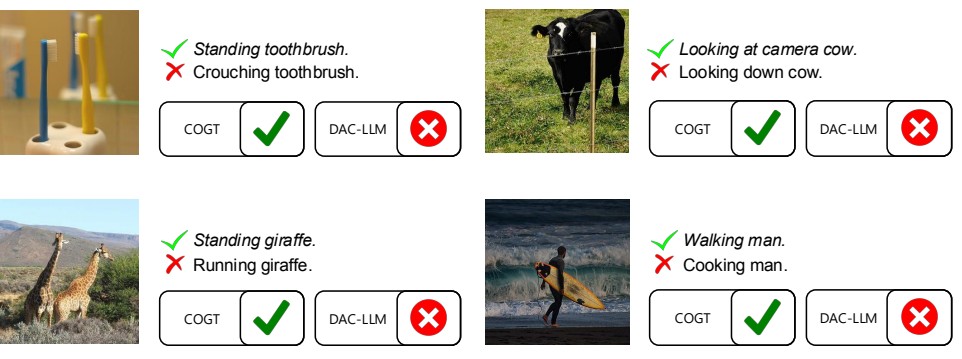

Figure 6: Qualitative results on sample images of VL-CheckList.

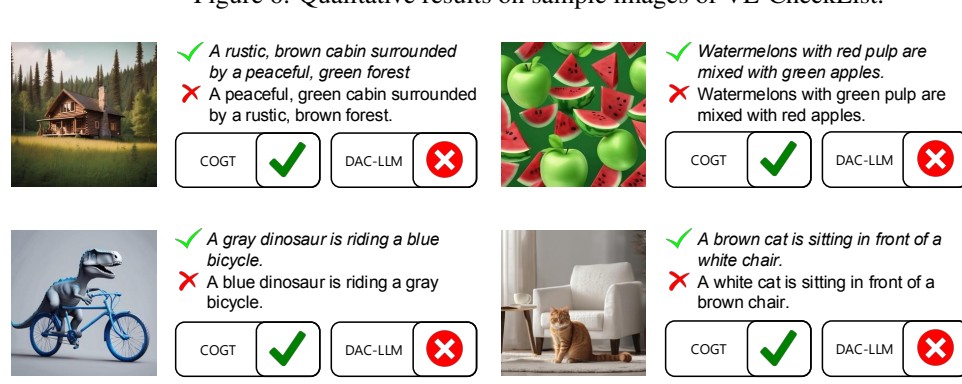

Figure 7: Qualitative results on sample images of ColorSwap.

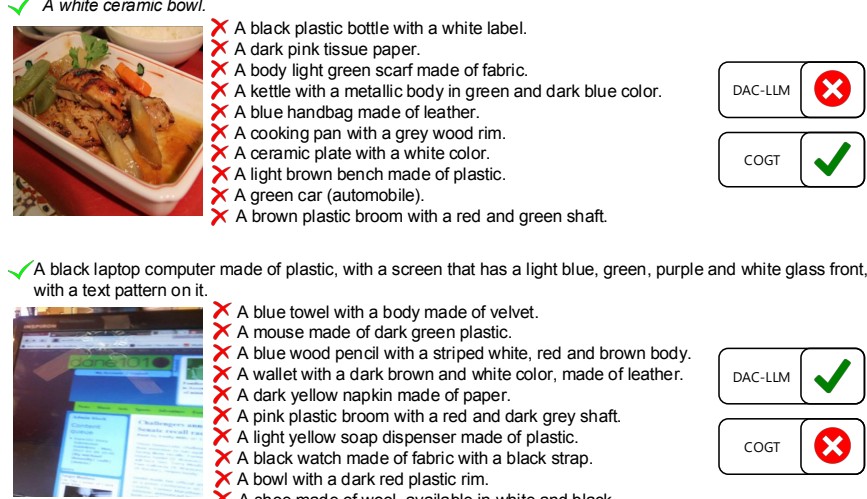

✓ *A white ceramic bowl.*

✗ A black plastic bottle with a white label.
✗ A dark pink tissue paper.
✗ A body light green scarf made of fabric.
✗ A kettle with a metallic body in green and dark blue color.
✗ A blue handbag made of leather.
✗ A cooking pan with a grey wood rim.
✗ A ceramic plate with a white color.
✗ A light brown bench made of plastic.
✗ A green car (automobile).
✗ A brown plastic broom with a red and green shaft.

DAC-LLM ❌
COGT ✓

✓ A black laptop computer made of plastic, with a screen that has a light blue, green, purple and white glass front, with a text pattern on it.

✗ A blue towel with a body made of velvet.
✗ A mouse made of dark green plastic.
✗ A blue wood pencil with a striped white, red and brown body.
✗ A wallet with a dark brown and white color, made of leather.
✗ A dark yellow napkin made of paper.
✗ A pink plastic broom with a red and dark grey shaft.
✗ A light yellow soap dispenser made of plastic.
✗ A black watch made of fabric with a black strap.
✗ A bowl with a dark red plastic rim.
✗ A shoe made of wool, available in white and black.

DAC-LLM ✓
COGT ❌

Figure 8: Qualitative results on sample images of the FG-OVD Trivial task. The image on the bottom shows a failure of COGT.

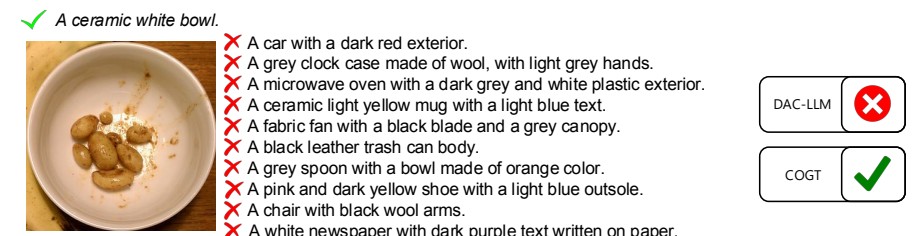

✓ *A ceramic white bowl.*

✗ A car with a dark red exterior.
✗ A grey clock case made of wool, with light grey hands.
✗ A microwave oven with a dark grey and white plastic exterior.
✗ A ceramic light yellow mug with a light blue text.
✗ A fabric fan with a black blade and a grey canopy.
✗ A black leather trash can body.
✗ A grey spoon with a bowl made of orange color.
✗ A pink and dark yellow shoe with a light blue outsole.
✗ A chair with black wool arms.
✗ A white newspaper with dark purple text written on paper.

DAC-LLM ❌
COGT ✓

Figure 9: Qualitative results on sample images of the FG-OVD Trivial task where COGT is successful while DAC-LLM fails.

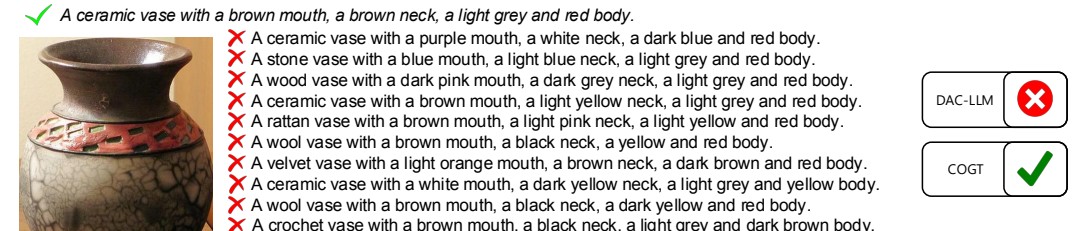

✓ *A ceramic vase with a brown mouth, a brown neck, a light grey and red body.*

✗ A ceramic vase with a purple mouth, a white neck, a dark blue and red body.
✗ A stone vase with a blue mouth, a light blue neck, a light grey and red body.
✗ A wood vase with a dark pink mouth, a dark grey neck, a light grey and red body.
✗ A ceramic vase with a brown mouth, a light yellow neck, a light grey and red body.
✗ A rattan vase with a brown mouth, a light pink neck, a light yellow and red body.
✗ A wool vase with a brown mouth, a black neck, a yellow and red body.
✗ A velvet vase with a light orange mouth, a brown neck, a dark brown and red body.
✗ A ceramic vase with a white mouth, a dark yellow neck, a light grey and yellow body.
✗ A wool vase with a brown mouth, a black neck, a dark yellow and red body.
✗ A crochet vase with a brown mouth, a black neck, a light grey and dark brown body.

DAC-LLM ❌
COGT ✓

Figure 10: Qualitative results on sample images of the FG-OVD Easy task.

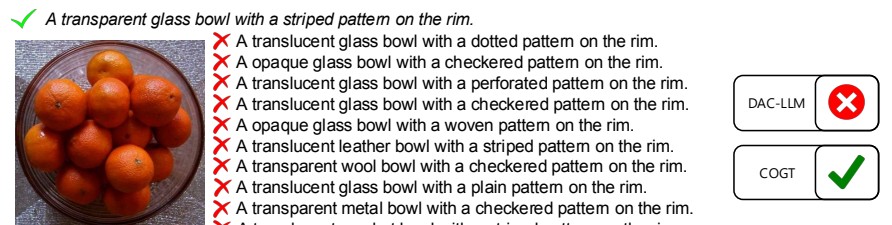

✓ *A transparent glass bowl with a striped pattern on the rim.*

✗ A translucent glass bowl with a dotted pattern on the rim.
✗ A opaque glass bowl with a checkered pattern on the rim.
✗ A translucent glass bowl with a perforated pattern on the rim.
✗ A translucent glass bowl with a checkered pattern on the rim.
✗ A opaque glass bowl with a woven pattern on the rim.
✗ A translucent leather bowl with a striped pattern on the rim.
✗ A transparent wool bowl with a checkered pattern on the rim.
✗ A translucent glass bowl with a plain pattern on the rim.
✗ A transparent metal bowl with a checkered pattern on the rim.
✗ A translucent crochet bowl with a striped pattern on the rim.

DAC-LLM ❌
COGT ✓

Figure 11: Qualitative results on sample images of the FG-OVD Medium task.

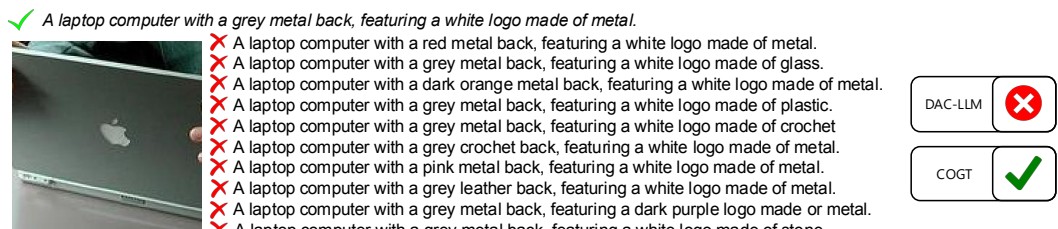

Figure 12: Qualitative results on sample images of the FG-OVD Hard task.

## F SYNTACTIC CATEGORIES

We report in Tab. 12 the 45 syntactic categories defined in (Silveira et al., 2014) and which form our set $V$ (Sec. 3).

Table 12: List of the syntactic categories defined in (Silveira et al., 2014).

| Syntactic Categories |
|---|
| acomp |
| advcl |
| advmod |
| amod |
| appos |
| aux |
| auxpass |
| cc |
| ccomp |
| conj |
| cop |
| csubj |
| csubjpass |
| dep |
| det |
| discourse |
| dobj |
| expl |
| goeswith |
| iobj |
| mark |
| mwe |
| neg |
| nn |
| npadvmod |
| nsubj |
| nsubjpass |
| num |
| number |
| parataxis |
| pcomp |
| pobj |
| poss |
| possessive |
| preconj |
| predet |
| prep |
| prt |
| punct |
| quantmod |
| rcmod |
| root |
| tmod |
| xcomp |

