# OpenReview forum: "Causal Graphical Models for Vision-Language Compositional Understanding"
_ICLR.cc/2025/Conference — ICLR 2025 Poster_

### Official Review · Reviewer_AYGs · 2024-10-30

**Soundness:** 3
**Presentation:** 3
**Contribution:** 3
**Rating:** 6
**Confidence:** 3

**Summary:**

The paper proposes to train a causal graphical model decoder on top of the 2-top-layer features extracted from a frozen pre-trained image encoder (clip / x-vlm). The decoder is used to "early-fusion" match between a textual caption parsed by a dependency parser and extracted image features. Trained on relatively small COCO train data following the NegCLIP paper protocol, the above small (38M params) decoder is able to achieve impressive image-text matching results on multiple compositional reasoning benchmarks, extending SOTA performance on those. Comparison is made to a variety of recent encoder-only methods (late fusion) showing significant gains. However, no comparison is made to modern LLM-decoder based LMMs that are known to also perform quite highly on the tested benchmarks (e.g. on SugarCREPE). That said, 38M params decoder is not apples-to-apples with a strong LLM :-)

**Strengths:**

- small training, small decoder, great results
- seems to be useful for CR checking, although not tested, can potentially aid in slow-fast type approaches to retrieval tasks (akin how original BLIP v1 did it) - might be a good idea to test, would make the contribution stronger I think
- contains adequate ablation

**Weaknesses:**

- no comparison to decoder LMMs (all the LLM-alignment llava-style methods) - those afaik perform strongly on those old CR benchmarks, especially with CapPa style inference, but multiple choice generation (answer with single letter, etc) also works well
- could test with CLIP-blind benchmark like eyes-wide-shut
- no applications beyond CR - eg to improve retrieval using eg slow-fast approach, where CLIP retrieves first and the proposed method (being early fusion it is heavier) filters
- It is not clear what 45 "standard syntactic categories" (line 288) are used in the LUT, I would expect a clear explanation on that in the paper and not sending to a reference. E.g. I would like to understand what the method is blind to? eg if some words get grouped to the same bucket etc...
- the write-up of the method is a bit convoluted and hard to follow, some details are omitted sending to prior work, ideally, the paper should be improved to make it more self-contained
- comparison to "methods trained on datasets larger then COCO" is not 100% fair both ways, as those datasets might also be much lower quality then COCO (eg synthetic or non-curated from the internet etc)

**Questions:**

see weaknesses

---

### Official Review · Reviewer_nWro · 2024-11-03

**Soundness:** 3
**Presentation:** 3
**Contribution:** 3
**Rating:** 6
**Confidence:** 4

**Summary:**

This paper introduces a method to improve vision-language compositional understanding by leveraging a dependency parser to construct a Dependency Tree (DT) for each sentence, which is then converted into a Causal Graphical Model (CGM). This CGM guides caption generation in a semi-parallel autoregressive approach, where each word is predicted based on its syntactic dependencies rather than a purely sequential order. They use CLIP as the vision encoder to extract visual features and train a decoder from scratch using the COCO dataset (additional experiments using X-VLM as the vision encoder are shown). This model is then used to validate their finetuning method -- Causally-Ordered Generative Training (COGT) -- on ARO, SugarCrepe, VL-Checklist, ColorSwap, and adapt FG-OVD to evaluate object attributes.

**Strengths:**

- This paper is easy to read, with extensive empirical results and useful qualitative examples.

- Motivation: using dependency/syntactic trees to capture the grammatical structure of sentences is well motivated -- this hierarchical
 structure shows how words depend on one another, which also may help to capture the compositional meaning of sentences. In vision-language models, capturing compositional meaning impacts on how different parts of a sentence relate to one another.

- This paper shows very strong results on 4 well known compositional benchmarks (ARO, SugarCrepe, VL-Checklist, ColorSwap), along with the annotated bounding boxes from the FG-OVD benchmark, and their positive and set of negative captions.

- The proposed method significantly outperforms prior works that explore captioning as the learning objective for image-text pair matching (Cap / CapPa) -- using a CLIP as the feature extractor for the visual features.

- The proposed approach to structuring predictions around syntactic dependencies allows the model to focus on learning relationships that are grammatically and semantically meaningful, reducing noise from irrelevant co-occurrences.

- Ablation results show results when varying different components (syntactic tree parser, masking by syntactic types, and using only the last layer or both the last and penultimate layer of CLIP for extracting the visual features)

**Weaknesses:**

- Generalization: One downside of this work is the overreliance on CLIP visual encoder as the feature extractor -- recent work has shown stronger performance using stronger baselines (e.g., average on SugarCrepe Swap -> CLIP acc. 64.25% vs. LLaVA acc. 81.25% -- using Contrastive Region Guidance [2] LLaVA acc. 90.75% -- results reported using CLIP+CGMs acc. is 83.14%). While X-VLM focuses on fine-grained alignment via cross-attention layers (for interactions between image and text features at multiple levels), it also falls short against high-performant VLMs such as LLaVA and InstructBLIP. For example, results using COGT compared to GPT-4V results on SugarCrepe are similar  -- it is not clear if using these backbones as feature extractors would yield better results:

| Model         | Add   | Replace | Swap  | Avg   |
|---------------|-------|---------|-------|-------|
| COGT + CLIP   | 98.26 | 87.10   | 83.14 | 89.50 |
| COGT + X-VLM  | **98.65** | 89.17   | 84.37 | **90.73** |
| GPT-4V       | 91.68 | **93.37**   | **86.61** | *90.55* |

- Missing prior work: in [3], authors show that using BLIP, arguably a very similar to CLIP backbone, and computing the match score of generating a particular text string given an image, yield SOTA results:

| Model            | ARO   |       | VL Checklist        |            |       | SugarCrepe                  |       |       |
|------------------|-------|-------|---------------------|-----------------------------------|-------|-------|-------|-------|
|                  | Rel.  | Attr. | Object             | Attribute                         | Rel.  | Replace | Swap  | Add   |
| COGT + CLIP      | 87.56 | 90.26 | 78.91              | **86.07**                             | 89.37 | 87.10  | 83.14 | 98.26 |
| COGT + X-VLM     | 87.64 | 92.30 | 80.49              | 85.87                             | 88.74 | 89.17  | 84.37 | **98.65** |
| VisualGPTScore [3]   | **89.1**  | **95.4**  | **94.4**               | 82.1                              | **92.8**  | **95.1**   | **92.4**  | 97.4  |

- Missing references: a) 3VL [1] is highly related to the proposed work: their method proposes to expand "the text of an arbitrary image-text pair into a hierarchical tree structure using language analysis tools, 3VL allows inducing this structure into the visual representation learned by the model, enhancing its interpretability and compositional reasoning" -- b) SynCLM [4], a syntax guided contrastive learning method, based on constituent and dependency structures of syntax trees, we propose phrase guided and tree-guided contrastive objectives for pre-training.

- Syntactic Trees and Causal Graphical Models analogy: while both STs and CGMs use directed acyclic graphs (DAGs) to represent relationships between elements, STs show linguistic hierarchy, while CGMs can show hierarchies of causal influence -- STs represent grammatical structure, while CGMs represent causal relationships; these are completely different domains with no inherent connection.
[5] identify these problems and propose to use causal grammars (as opposed to linguistic grammars, as suggested in this work). In the context of this work, STs represent grammatical relationships, but it's hard to regard them as causal ones.
As an example, syntactic dependencies represent grammatical relationships between words in a sentence. For example, in the sentence "a brown bird has a small yellow head," the dependency tree shows that "bird" is the head of "brown," and "head" is the head of "small yellow." -- these dependencies are linguistic in nature, indicating how words relate to each other grammatically -- here, "bird" being the head of "brown" doesn't imply that "bird" causes "brown" to exist; it simply means that "brown" modifies "bird" grammatically.

- Disentangled Factorization and Syntactic Trees: the relationships in syntactic trees do not imply that one word causes another word to appear or behave in a certain way: they describe how words are structured grammatically. Treating syntactic dependencies as causal dependencies is conceptually difficult to correlate because the nature of the relationships is fundamentally different -- the joint distribution of words in a sentence (Eq 4) reduces the complexity of modeling by focusing only on the dependencies indicated by the syntactic tree. However, the factorization is based on grammatical dependencies, not causal dependencies.

- An important aspect of causal reasoning is the ability to perform interventions (using Pearl's do-calculus) or reason about counterfactuals (what would happen if something were different). In COGT, there is no notion of intervening in one word to see how it would change another word's appearance or meaning -- instead, the model predicts words based on their syntactic dependencies without considering how one word might causally influence another through an intervention.


[1]  Yellinek, Nir, Leonid Karlinsky, and Raja Giryes. "3VL: using Trees to teach Vision & Language models compositional concepts." arXiv preprint arXiv:2312.17345 (2023).

[2] Wan, David, et al. "Contrastive region guidance: Improving grounding in vision-language models without training." arXiv preprint arXiv:2403.02325 (2024).

[3] Lin, Zhiqiu et al. “Revisiting the Role of Language Priors in Vision-Language Models.” International Conference on Machine Learning (2023).

[4] Zhang, Shuai et al. “Syntax-guided Contrastive Learning for Pre-trained Language Model.” Findings (2022).

[5] Tenenbaum, J. B., Griffiths, T. L., & Niyogi, S. (2007). Intuitive theories as grammars for causal inference. In A. Gopnik & L. Schulz (Eds.), Causal learning: Psychology, philosophy, and computation (pp. 301–322). Oxford University Press. https://doi.org/10.1093/acprof:oso/9780195176803.003.0020

**Questions:**

- Is it possible to use better performant backbones (e.g., LLaVA, InstructBLIP, BLIP2) and finetune the model rather than train an independent module?

- What is the difference between Deep Biaffine (alone) vs Deep Biaffine + RoBERTa? Is this related to the discrepancy of the dependency parser between word-level vs sub-word tokenization to build the dependency tree?

- The failure case shown in Figure 5 (Supp Material) opens an interesting question: for long sentences, which are usually not present in the evaluated benchmarks, there are multiple layers of dependencies (e.g., adjectives modifying nouns, prepositional phrases providing additional information, relative clauses, etc). In these cases, each word might have multiple ancestors in the syntactic tree (e.g., "light blue" depends on "screen," which depends on "laptop"), leading to a more complex factorization; it can be indeed difficult to parse or resolve correctly. More importantly, it may become even more apparent that these relationships are not causal: for example, "a black laptop computer made of plastic" does not cause "a screen that has a light blue glass front." -- these are separate descriptions that happen to be part of the same sentence. May be this the reason why the model fails or struggles in this case?

- In complex sentences, for example, "A black laptop computer made of plastic, with a screen that has a light blue, green, purple and white glass front, with a text pattern on it."; "light blue" depends on both "screen" and indirectly on "laptop." -- given sentences with multiple hierarchical dependencies, how are these factors balanced? It seems that with more dependencies (e.g., through conjunctions or relative clauses), maintaining accurate conditional independence assumptions becomes harder.

______

- Friendly suggestion (not to pay particular attention): there is interesting related work [6, 7] that relies on graph-based representations to model relationships between entities in sentences and scenes (both language and visual inputs) -- using scene graphs as a structured representation of an image seems orthogonal to the proposed approach in this paper.

- I'm also genuinely curious about the performance of COGT on Winoground -- given the ambiguity of some sentences, it is not particularly clear how finetuning using the structure yielded via STs would impact when evaluating this benchmark.

[6] Singh, Harman et al. “Coarse-to-Fine Contrastive Learning in Image-Text-Graph Space for Improved Vision-Language Compositionality.” Conference on Empirical Methods in Natural Language Processing (2023).

[7] Herzig, Roei et al. “Incorporating Structured Representations into Pretrained Vision & Language Models Using Scene Graphs.” ArXiv abs/2305.06343 (2023): n. pag.

---

> ### Comment · Reviewer_nWro · 2024-11-27
>
> Thank you so much for such a thoughtful response and going over every concern and question. I really appreciate the analogies and explanations introduced in the rebuttal and revised paper to help justify the motivation behind the proposed work in the current form; I think this allows for better understanding and ground for future work.
>
> However, in light of new experimental evidence, I have additional concerns:
>
> *"Note that we significantly outperform these results with both COGT-InstructBLIP+ and COGT-XVLM+"*
>
> However, unfortunately, this claim does not hold and falls when comparing against BLIP-2 (a model that is consistently outperformed by InstructBLIP in several benchmarks [1]) - Table 9 of [2]:
>
> |                | COGT-InstructBLIP+ | VisualGPTScore (α=0 w/ BLIP) | Difference |
> |----------------|---------------------|-----------------------------|------------|
> | Aro           | 92.2               | 93.02                       | +0.82    |
> | Sugar Crepe   | 91.77              | 92.11                       | +0.34    |
> | VL-Checklist  | 87.37              | 89.63                       | +2.26    |
>
>
> |                | COGT-InstructBLIP+ | VisualGPTScore (α=0 w/ BLIP-2) | Difference |
> |----------------|---------------------|--------------------------------|------------|
> | Aro           | 92.2               | 92.5                          | -0.3     |
> | Sugar Crepe   | 91.77              | 92.31                         | -0.54    |
> | VL-Checklist  | 87.37              | 84.33                         | +3.04    |
>
> As a reminder, VisualGPTScore with BLIP-2 (α=0) is not re-trained, and yet, outperforms COGT-InstructBLIP+ in almost all evaluated benchmarks.
>
> Another concern revolves around Winoground. It is unfortunate that one prior published work has dismissed Winoground as a strong baseline. This, is by far, one of the most challenging and evaluated benchmark for compositionality as of today. For example, **https://paperswithcode.com/sota/visual-reasoning-on-winoground** contains 104 results in this benchmark, and this number is constantly growing. However, as of today, SugarCrepe is only tested in a few methods, with a significantly lower number of works focusing on it.
>
> Moreover, [3] reports group scores of 21.3 using BLIP-2 Score, and 28.5 using InstructBLIP with vqaScore [3]. These numbers significantly outperform the reported using COGT+InstructBLIP+. More specifically:
>
> |                | COGT+InstructBLIP+ | VQA Score (w/ InstructBLIP) | Difference |
> |----------------|---------------------|-------------------------------------------|------------|
> | Winoground    | 13                  | 28.5                                      |  -15.5    |
> | | | | |
>
> I strongly ask the authors to include Winoground (which is broadly used and commonly reported), and to downtone their claims on outperforming the SOTA, as their work shows competitive results against prior strong baselines. With this, I'm keeping my original rating.
>
> _______
>
> [1] Dai, Wenliang et al. "InstructBLIP: Towards General-purpose Vision-Language Models with Instruction Tuning." In Thirty-seventh Conference on Neural Information Processing Systems (2023).
>
> [2] Lin, Zhiqiu et al. “Revisiting the Role of Language Priors in Vision-Language Models.” International Conference on Machine Learning (2023).
>
> [3] Lin, Zhiqiu et al. “Evaluating Text-to-Visual Generation with Image-to-Text Generation.” European Conference on Computer Vision (2024).

---

### Official Review · Reviewer_tN1p · 2024-11-04

**Soundness:** 4
**Presentation:** 4
**Contribution:** 3
**Rating:** 8
**Confidence:** 3

**Summary:**

This paper models the dependency relations among textual and visual tokens using a causal graphical model (CGM) built using a dependency parser. The proposed dependency-constraint decoding follows the CGM structure which encourages a focus on the main causal dependencies. On several vision-language compositional benchmarks, the proposed method archives significant performance gains over baselines.

**Strengths:**

1. The author propose a causal graphical model with dependency-constraint decoding method to incorporate compositional knowledge into VLM. Using relationship representations could effectively model the compositionally of the data and resolve the long-existing issue for sequential modeling VLM that the model treats data as bag-of-words.
2. Experiments on several vision-language compositional benchmarks show that the proposed method is effective in improving the compositional performance. Specifically, the proposed method outperforms all other baselines trained on COCO only by a large margin.

**Weaknesses:**

1. The training objective might not be robust to mislabeled data, thus posing challenging to strict data cleaning when future work consider scaling up.
2. The author did not compare the efficiency in terms of training nor inference of the proposed method with other baselines.
3. The evaluation focuses on retrieval, where the model is tasked to find the correct caption for a given image and vice versa. It remains unclear how to improve the compositional generation ability, which is of a greater need in real-world application.

**Questions:**

1. How is the dependency corpus by Silveira et al. representative in web-scale data? Do the authors think future work should extend it or it is enough for future applications?
2. How does the dependency parser handle cross-boundary entities? For example, there might exist data that refers to a previous entity that cannot be interpreted without the full context.
3. How could the dependency-constrained decoding extend to generation tasks (i.e., evaluated on non-retrieval task format)? Would the constrained decoding harm the language modeling quality?
4. What could be potential ways to address the challenge that CLIP backbone cannot detect out-of-focus objects for compositional reasoning?
5. When scaling up, how do the author handle falsely annotated noisy data in the web-scale data pool?

**Details Of Ethics Concerns:**

No concern.

---

### Meta-Review · Area_Chair_wS5c · 2024-12-09

**Metareview:**

This paper presents a new method to improve vision-language (VL) compositional understanding by leveraging a dependency parser to construct a Dependency Tree (DT) for each sentence, which is then converted into a Causal Graphical Model (CGM). The proposed method achieves great results o several VL compositional benchmarks. After rebuttal, it received scores of 668, and all the reviewers are happy about the paper, commenting that (1) the proposed method is solid, specifically, the analogies and intuitions to connect syntactic trees and CGMs are proved to be helpful; (2) the paper contains adequate ablation and results are strong on 4 well known compositional benchmarks. Therefore, the AC would like to recommend acceptance of the paper.

**Additional Comments On Reviewer Discussion:**

After rebuttal, all the reviewers decided to keep their positive scores. Specifically,

1. Reviewers have asked additional results using LLaVA-like backbones. During rebuttal, the authors have provided additional results using InstructBLIP as the backbone.

2. Reviewers have asked to perform larger-scale model training beyond COCO. During rebuttal, the authors have provided additional results using CC3M.

3. Reviewers have asked additional results on Winoground and testing on eyes-wide-shut. During rebuttal, the authors have provided additional results on these benchmarks.

4. The authors have also done a reasonably good job on dealing the concern how to extend the method to non-retrieval tasks.

5. Reviewers have asked additional questions regarding missing references, analogies between Dependency Trees and CGMs, etc., and the authors have done a nice job.

---

### Decision · Program_Chairs · 2025-01-22

Accept (Poster)